# ON THE ADVERSARIAL ROBUSTNESS OF 3D POINT CLOUD CLASSIFICATION

## ABSTRACT

3D point clouds play pivotal roles in various safety-critical fields, such as autonomous driving, which desires the corresponding deep neural networks to be robust to adversarial perturbations. Though a few defenses against adversarial point cloud classification have been proposed, it remains unknown whether they can provide *real* robustness. To this end, we perform the first security analysis of state-of-the-art defenses and design adaptive attacks on them. Our 100% adaptive attack success rates demonstrate that current defense designs are still vulnerable. Since adversarial training (AT) is believed to be the most effective defense, we present the first in-depth study showing how AT behaves in point cloud classification and identify that the required symmetric function (pooling operation) is paramount to the model's robustness under AT. Through our systematic analysis, we find that the default used fixed pooling operations (*e.g.,* `MAX` pooling) generally weaken AT's performance in point cloud classification. Still, *sorting-based* parametric pooling operations can significantly improve the models' robustness. Based on the above insights, we further propose `DeepSym`, a deep symmetric pooling operation, to architecturally advance the adversarial robustness under AT to 47.0% without sacrificing nominal accuracy, outperforming the original design and a strong baseline by 28.5% ($\sim 2.6\times$) and 6.5%, respectively, in PointNet.

## 1 INTRODUCTION

Despite the prominent achievements that deep neural networks (DNN) have reached in the past decade, adversarial attacks (Szegedy et al., 2013) are becoming the Achilles' heel in modern deep learning deployments, where adversaries generate imperceptible perturbations to mislead the DNN models. Numerous attacks have been deployed in various 2D vision tasks, such as classification (Carlini & Wagner, 2017), object detection (Song et al., 2018), and segmentation (Xie et al., 2017). Since adversarial robustness is a critical feature, tremendous efforts have been devoted to defending against 2D adversarial images (Guo et al., 2017; Papernot et al., 2016; Madry et al., 2018). However, Athalye et al. (2018) suggest that most of the current countermeasures essentially try to obfuscate gradients, which give a false sense of security. Besides, certified methods (Zhang et al., 2019) often provide a lower bound of robustness, which are not helpful in practice. Therefore, adversarial training is widely believed as the most and only effective defense solution.

The emergence of 3D point cloud applications in safety-critical areas like autonomous driving raises public concerns about their security of DNN pipelines. A few studies (Xiang et al., 2019; Cao et al., 2019; Sun et al., 2020) have demonstrated that various deep learning tasks on point clouds are indeed vulnerable to adversarial examples. Among them, point cloud classification models have laid solid foundations upon which other complex models are built (Lang et al., 2019; Yu et al., 2018a). While it seems intuitive to extend convolutional neural networks (CNN) from 2D to 3D for point cloud classification, it is actually not a trivial task. The difficulty mainly inherits from that point cloud is an *unordered* set structure that CNN cannot handle. Modern point cloud classification models (Qi et al., 2017a; Zaheer et al., 2017) address this problem by leveraging a **symmetric function**, which is *permutation-invariant* to the order of points, to aggregate local features, as shown in Figure 2.

Recently, a number of countermeasures have been proposed to defend against 3D adversarial point clouds. However, the failure of gradient obfuscation-based defenses in the 2D space motivates us to re-think whether current defense designs provide *real* robustness for 3D point cloud classification. Especially, DUP-Net (Zhou et al., 2019) and GvG-PointNet++ (Dong et al., 2020a) claim to improve the adversarial robustness significantly. However, we find that both defenses belong to gradient

obfuscation through our analysis, hence further design white-box adaptive attacks to break their robustness. Unfortunately, our 100% attack success rates demonstrate that current defense designs are still vulnerable.

As mentioned above, adversarial training (AT) is considered the most effective defense strategy; we thus perform the *first* rigorous study of how AT behaves in point cloud classification by exploiting projected gradient descent (PGD) attacks (Madry et al., 2018). We identify that the default used **symmetric function** weakens the effectiveness of AT. Specifically, popular models (*e.g.,* PointNet) utilize fixed pooling operations like MAX and SUM pooling as their symmetric functions to aggregate features. Different from CNN-based models that usually apply pooling operations with a small sliding window (*e.g.,* $2 \times 2$), point cloud classification models leverage such fixed pooling operations to aggregate features from a large number of candidates (*e.g.,* 1024). We find that those fixed pooling operations inherently lack flexibility and learnability, which are not appreciated by AT. Moreover, recent research has also presented parametric pooling operations in set learning (Wang et al., 2020; Zhang et al., 2020), which also preserve permutation-invariance. We take a step further to systematically analyze point cloud classification models' robustness with parametric pooling operations under AT. Experimental results show that the *sorting-based* pooling design benefits AT well, which vastly outperforms MAX pooling, for instance, in adversarial accuracy by 7.3% without hurting the nominal accuracy[1].

Lastly, based on our experimental insights, we propose DeepSym, a sorting-based pooling operation that employs deep learnable layers, to architecturally advance the adversarial robustness of point cloud classification models under AT. Experimental results show that DeepSym reaches the best adversarial accuracy in all chosen backbones, which on average, is a 10.8% improvement compared to the default architectures. We also explore the limits of DeepSym based on PointNet due to its broad adoption (Guo et al., 2020). We obtain the best robustness on ModelNet40, which achieves the adversarial accuracy of 47.0%, significantly outperforming the default MAX pooling design by 28.5% ($\sim 2.6\times$). In addition, we demonstrate that PointNet with DeepSym also reaches the best adversarial accuracy of 45.2% under the most efficient AT on ModelNet10 (Wu et al., 2015), exceeding MAX pooling by 17.9% ($\sim 1.7\times$).

## 2 BACKGROUND AND RELATED WORK

**3D point cloud classification.** Early works attempt to classify point clouds by adapting deep learning models in the 2D space (Su et al., 2015; Yu et al., 2018b). DeepSets (Zaheer et al., 2017) and PointNet (Qi et al., 2017a) are the first to achieve end-to-end learning on point cloud classification and formulate a general specification (Figure 2) for point cloud learning. PointNet++ (Qi et al., 2017b) and DGCNN (Wang et al., 2019) build upon PointNet set abstraction to better learn local features. Lately, DSS (Maron et al., 2020) generalizes DeepSets to enable complex functions in set learning. Besides, ModelNet40 (Wu et al., 2015) is the most popular dataset for benchmarking point cloud classification, which consists of 12,311 CAD models belonging to 40 categories. The numerical range of the point cloud data is normalized to $[-1, 1]$ in ModelNet40.

**Adversarial attacks and defenses on point clouds.** Xiang et al. (2019) perform the first study to extend C&W attack (Carlini & Wagner, 2017) to point cloud classification. Wen et al. (2019) improve the loss function in C&W attack to realize attacks with smaller perturbations and Hamdi et al. (2019) present black-box attacks on point cloud classification. Recently, Zhou et al. (2019) and Dong et al. (2020a) propose to defend against adversarial point clouds by input transformation and adversarial detection. Besides, Liu et al. (2019) conduct a preliminary investigation on extending countermeasures in the 2D space to defend against simple attacks like FGSM (Goodfellow et al., 2014) on point cloud data. In this work, we first design adaptive attacks to break existing defenses and analyze the adversarial robustness of point cloud classification under adversarial training.

## 3 BREAKING THE ROBUSTNESS OF EXISTING DEFENSES

### 3.1 ADAPTIVE ATTACKS ON DUP-NET

**DUP-Net** (ICCV'19) presents a denoiser layer and upsampler network structure to defend against adversarial point cloud classification. The denoiser layer $g : \mathbb{X} \to \mathbb{X}'$ leverages $k$NN ($k$-nearest

---

[1] In this paper, we use nominal and adversarial accuracy to denote the model's accuracy on clean and adversarially perturbed data, respectively.

neighbour) for outlier removal. Specifically, the $k$NN of each point $\boldsymbol{x}_i$ in point cloud $\mathbb{X}$ is defined as $knn(\boldsymbol{x}_i, k)$ so that the average distance $d_i$ of each point $\boldsymbol{x}_i$ to its $k$NN is denoted as:

$$d_i = \frac{1}{k} \sum_{\boldsymbol{x}_j \in knn(\boldsymbol{x}_i, k)} ||\boldsymbol{x}_i - \boldsymbol{x}_j||_2 \ , \quad i = \{1, 2, \ldots, n\} \tag{1}$$

where $n$ is the number of points. The mean $\mu = \frac{1}{n} \sum_{i=1}^n d_i$ and standard deviation $\sigma = \sqrt{\frac{1}{n} \sum_{i=1}^n (d_i - \mu)^2}$ of all these distances are computed to determine a distance threshold as $\mu + \alpha \cdot \sigma$ to trim the point clouds, where $\alpha$ is a hyper-parameter. As a result, the denoised point cloud is represented as $\mathbb{X}' = \{\boldsymbol{x}_i \mid d_i < \mu + \alpha \cdot \sigma\}$. The denoised point cloud $\mathbb{X}'$ will be further fed into PU-Net (Yu et al., 2018a), defined as $p : \mathbb{X}' \to \mathbb{X}''$, to upsample $\mathbb{X}'$ to a fixed number of points. Combined with the classifier $f$, the integrated DUP-Net can be noted as $(f \circ p \circ g)(\mathbb{X})$. The hypothesis is that the denoiser layer will eliminate the adversarial perturbations and the upsampler network will re-project the denoised off-manifold point cloud to the natural manifold.

**Analysis.** The upsampler network $p$ (*i.e.,* PU-Net) is differentiable and can be integrated with the classification network $f$. Therefore, $f \circ p$ is clearly vulnerable to gradient-based adaptive attacks. Although the denoiser layer $g$ is not differentiable, it can be treated as *deterministic masking*: $\mathcal{M}(\boldsymbol{x}_i) = \mathbf{1}_{d_i < \mu + \alpha \cdot \sigma}$ so that the gradients can still flow through the masked points. By involving $\mathcal{M}(\boldsymbol{x}_i)$ into the iterative optimization process: $\nabla_{\boldsymbol{x}_i}(f \circ p \circ g)(\mathbb{X})|_{\boldsymbol{x}_i = \hat{\boldsymbol{x}}} \approx \nabla_{\boldsymbol{x}_i}(f \circ p)(\mathbb{X})|_{\boldsymbol{x}_i = \hat{\boldsymbol{x}} \cdot \mathcal{M}(\hat{\boldsymbol{x}})}$, similar to BPDA (Athalye et al., 2018), attackers may still find adversarial examples.

**Experimentation.** We leverage the open-sourced codebase[2] of DUP-Net for experimentation. Specifically, a PointNet (Qi et al., 2017a) trained on ModelNet40 is used as the target classifier $f$. For the PU-Net, the upsampled number of points is 2048, and the upsampling ratio is 2. For the adaptive attacks, we exploit targeted $L^2$ norm-based C&W attack and untargeted $L^\infty$ norm-based PGD attack with 200 iterations (PGD-200). Detailed setups are elaborated in Appendix A.1.

Table 1: Adversarial accuracy under adaptive attacks on PU-Net and DUP-Net. For the denoiser layer $g$, $k = 2$ and $\alpha = 1.1$ are set the same as Zhou et al. (2019). † denotes the attack in the original paper.

| Attack Method | Adversarial Accuracy | | | Mean $L^2$ Norm Distance |
| :---: | :---: | :---: | :---: | :---: |
| | PointNet ($f$) | PU-Net ($f \circ p$) | DUP-Net ($f \circ p \circ g$) | |
| Clean point cloud | 88.3% | 87.5% | 86.3% | 0.0 |
| C&W attack on $f$ † | **0.0%** | 23.9% | 84.5% | 0.77 |
| C&W attack on $f \circ p$ | 2.3% | **0.0%** | 74.7% | 0.71 |
| Adaptive attack on $f \circ p \circ g$ | 1.1% | 0.8% | **0.0%** | 1.62 |
| PGD attack ($\epsilon = 0.01$) | 7.1% | 5.9% | 5.4% | - |
| PGD attack ($\epsilon = 0.025$) | 3.5% | 2.8% | 2.1% | - |
| PGD attack ($\epsilon = 0.05$) | 1.3% | 1.0% | 0.8% | - |
| PGD attack ($\epsilon = 0.075$) | 0.0% | 0.0% | **0.0%** | - |

**Discussion.** As shown in Table 1, adaptive C&W attacks achieve 100% success rates on DUP-Net. Though the mean distance of adversarial examples targeting DUP-Net is larger than those targeting PU-Net, they are almost indistinguishable by human perception, as visualized in Appendix A.2. We find that naïve PGD attacks are also effective since the upsampler network is sensitive to $L^\infty$ norm-based perturbations. The design of DUP-Net is similar to ME-Net (Yang et al., 2019) in the 2D space, which recently has been shown vulnerable to adaptive attacks (Tramer et al., 2020). We hereby demonstrate that such input transformation-based defenses cannot offer real robustness to point cloud classification, either.

## 3.2 Adaptive Attacks on GvG-PointNet++

**GvG-PointNet++** (CVPR'20) introduces gather vectors in the 3D point clouds as an adversarial indicator. The original PointNet++ aggregates local features $\boldsymbol{f}_i$ hierarchically to make final classification. Gather vectors $\boldsymbol{v}_i$ are learned from local features $\boldsymbol{f}_i$ to indicate the global center $\boldsymbol{c}_i$ of a point cloud sample. If the indicated global center $\boldsymbol{c}_i$ is far away from the ground-truth global center $\boldsymbol{c}_g$, the corresponding local feature $\boldsymbol{f}_i$ will be masked out:

$$\boldsymbol{c}_i = \boldsymbol{x}_{ci} + \boldsymbol{v}_i \ ; \quad \mathcal{M}_i = \mathbf{1}_{||\boldsymbol{c}_g - \boldsymbol{c}_i|| < r} \ ; \quad \mathbb{F}_g = \{\boldsymbol{f}_i \cdot \mathcal{M}_i\} \tag{2}$$

---

[2] https://github.com/RyanHangZhou/DUP-Net

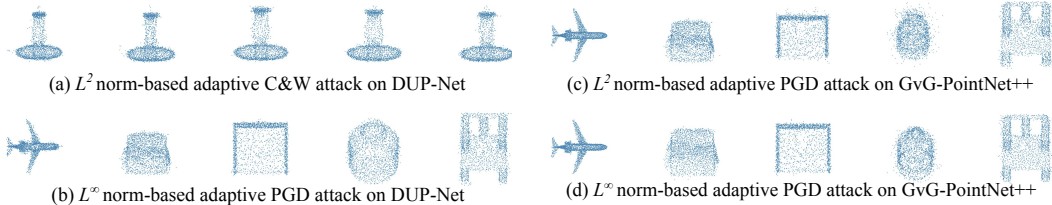

(a) $L^2$ norm-based adaptive C&W attack on DUP-Net

(c) $L^2$ norm-based adaptive PGD attack on GvG-PointNet++

(b) $L^\infty$ norm-based adaptive PGD attack on DUP-Net

(d) $L^\infty$ norm-based adaptive PGD attack on GvG-PointNet++

Figure 1: Sampled visualizations of adversarial examples generated by adaptive attacks on both defenses ($\epsilon = 0.05$ and $\delta = 0.16$). More visualizations can be found in Appendix A.2.

where $\boldsymbol{x}_{ci}$ is the geometry center of the local point set, $r$ is the distance threshold to mask the local feature, and $\mathbb{F}_g$ is the cleaned feature set for final classification. To train GvG-PointNet++, it is necessary to optimize a surrogate loss to correctly learn the gather vectors besides the cross-entropy (xent) loss:

$$\mathcal{L}_{total} = \mathcal{L}_{xent} + \lambda \cdot \mathcal{L}_{gather} \ , \quad \mathcal{L}_{gather} = \sum_{i=1}^{n'} ||\boldsymbol{c}_i - \boldsymbol{c}_g||_1 \qquad (3)$$

where $n'$ is the number of local features and $\lambda$ is a hyper-parameter. Thus, GvG-PointNet++ essentially applies self-attention to the local features and relies on it for robustness enhancement.

**Analysis.** Dong et al. (2020a) evaluate white-box adversaries on GvG-PointNet++ with naïve $L^2$ norm-based PGD attacks. Specifically, only $\mathcal{L}_{xent}$ is utilized in the adversarial optimization process so that the masking $\mathcal{M}_i$ will hinder the gradient propagation. However, since $\mathcal{M}_i$ is learned from the network itself, it is highly possible to further break this obfuscation with $\mathcal{L}_{gather}$ considered. The adaptive attack can be then formulated as an optimization problem with the loss function:

$$\mathcal{L}_{adv} = \mathcal{L}_{xent} - \beta \cdot \mathcal{L}_{gather} \qquad (4)$$

where $\beta$ is a hyper-parameter. By maximizing $\mathcal{L}_{adv}$ with $L^2$ norm-based PGD attacks, adversaries strive to enlarge the adversarial effect but also minimize the perturbations on gather vectors. We also find that GvG-PointNet++ is by design vulnerable to PGD attacks on $\mathcal{L}_{gather}$ as such perturbations will potentially affect most gather vector predictions to make $\boldsymbol{g}_i$ masked out so that insufficient for final classification.

**Experimentation.** We train GvG-PointNet++ based on single-scale grouped PointNet++ (Qi et al., 2017b) on ModelNet40 and set $r = 0.08$ and $\lambda = 1$ as suggested by Dong et al. (2020a). The model is trained by Adam (Kingma & Ba, 2014) optimizer with 250 epochs using batch size 16, and the initial learning rate is 0.01. For the adaptive attack, we use 10-step binary search to find a appropriate $\beta$. The setup of $L^2$ norm-based PGD attacks is identical to Dong et al. (2020a), and we also leverage $L^\infty$ norm-based PGD-200 in the evaluation. Detailed setups are elaborated in Appendix A.1.

Table 2: Adversarial accuracy under $L^p$ norm-based adaptive attacks on GvG-PointNet++. $\epsilon$ and $\delta$ are the perturbation boundaries. † denotes the attack in the original paper.

| Target Loss | Adversarial Accuracy ($L^\infty$) | | | | Adversarial Accuracy ($L^2$) | | |
|---|---|---|---|---|---|---|---|
| | $\epsilon = 0.01$ | $\epsilon = 0.025$ | $\epsilon = 0.05$ | $\epsilon = 0.075$ | $\delta = 0.08$ | $\delta = 0.16$ | $\delta = 0.32$ |
| $\mathcal{L}_{xent}$† | 30.6% | 21.4% | 5.4% | 1.8% | 25.2% | 16.9% | 15.4% |
| $\mathcal{L}_{adv}$ | 20.1% | 12.6% | 2.2% | **0.0%** | **7.5%** | 4.4% | **2.1%** |
| $\mathcal{L}_{gather}$ | **17.9%** | **8.1%** | **0.0%** | **0.0%** | 8.5% | **4.1%** | 2.7% |

**Discussion.** As shown in Table 2, both adaptive PGD attacks achieve high success rates on GvG-PointNet++. we also observe that the $L^\infty$ norm-based PGD attack is more effective on $\mathcal{L}_{gather}$ since $L^\infty$ norm perturbations assign the same adversarial budget to each point, which can easily impact a large number of gather vector predictions. However, it is hard for the $L^2$ norm-based PGD attack to influence so many gather vector predictions because it prefers to perturb key points rather than the whole point set. GvG-PointNet++ leverages DNN to detect adversarial perturbations, which is similar to MagNet (Meng & Chen, 2017) in the 2D space. We validate that adversarial detection also fails to provide real robustness under adaptive white-box adversaries in point cloud classification.

## 4 ADVERSARIAL TRAINING WITH DIFFERENT SYMMETRIC FUNCTIONS

We have so far demonstrated that state-of-the-art defenses against 3D adversarial point clouds are still vulnerable to adaptive attacks. While gradient obfuscation cannot offer real adversarial robust-

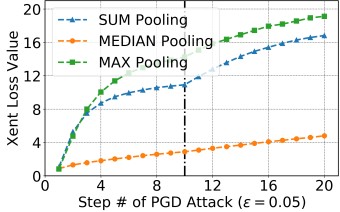

Figure 2: The general specification of point cloud classification ($\sigma \circ \rho \circ \Phi)(\mathbb{X})$, where $n$ is the number of points, $d_i$ is the number of hidden dimensions in the $i$-th feature map, $\Phi(\cdot)$ represents the permutation-equivariant layers, $\rho(\cdot)$ denotes the column-wise symmetric (permutation-invariant) function, and $\sigma(\cdot)$ is the fully connected layer.

Figure 3: Xent loss of PGD attack on PointNet with three fixed pooling operations (each value is averaged over 100 runs from random starting points).

ness, adversarial training (AT) is widely believed to be the most effective method. In this section, we conduct the first thorough study showing how AT performs in point cloud classification.

## 4.1 ADVERSARIAL TRAINING PRELIMINARIES

Madry et al. (2018) formulate AT as a paddle point problem in Equation 5, where $\mathcal{D}$ is the underlying data distribution, $\mathcal{L}(\cdot, \cdot, \cdot)$ is the loss function, $x$ is the training data with its label $y$, $\epsilon$ is the adversarial perturbation, and $\mathbb{S}$ denotes the boundary of such perturbations.

$$\arg \min_{\theta} \quad \mathbb{E}_{(x,y)\sim\mathcal{D}} \left[ \max_{\epsilon \in \mathbb{S}} \mathcal{L}(x + \epsilon, y, \theta) \right] \tag{5}$$

**Adversarial training setups.** We choose PointNet (Qi et al., 2017a), DeepSets (Zaheer et al., 2017), and DSS (Maron et al., 2020) as the backbone networks. As shown in Section 3 and demonstrated by Madry et al. (2018), $L^\infty$ norm-based PGD attack tends to be a universal first-order adversary. Thus, we select PGD-7 into the training recipe, and empirically set the maximum per-point perturbation $\epsilon = 0.05$ out of the point cloud range $[-1, 1]$. We follow the default PointNet training setting to train models on the ModelNet40 training set. In the evaluation, we utilize PGD-200 to assess their robustness on the ModelNet40 validation set with the same adversarial budget $\epsilon = 0.05$. Meanwhile, we also report the nominal accuracy on the clean validation set. Each PGD attack starts from a random point in the allowed perturbation space. More details can be found in Appendix B.

## 4.2 ADVERSARIAL TRAINING WITH FIXED POOLING OPERATIONS

As shown in Figure 2, modern models fundamentally follow a general specification $(\sigma \circ \rho \circ \Phi)(\mathbb{X})$ for point cloud classification. $\Phi(\cdot)$ represents a set of permutation-equivariant layers to learn local features from each point. $\rho(\cdot)$ is a column-wise symmetric (permutation-invariant) function to aggregate the learned local features into a global feature, and $\sigma(\cdot)$ are fully connected layers for final classification. PointNet, DeepSets, and DSS leverage different $\Phi(\cdot)$ for local feature learning, but all depend on **fixed pooling operations** as their $\rho(\cdot)$. Specifically, MAX pooling is by default used in DeepSets (for point cloud classification) and PointNet (Zaheer et al., 2017; Qi et al., 2017a), while DSS utilizes SUM pooling (Maron et al., 2020). We also additionally select MEDIAN pooling due to its robust statistic feature (Huber, 2004). Though models with fixed pooling operations have achieved satisfactory accuracy under standard training, they face various difficulties under AT. As shown in Table 3, models with MEDIAN pooling achieve better nominal accuracy among fixed pooling operations, but much worse adversarial accuracy, while SUM pooling performs contrarily. Most importantly, none of them reach a decent balance of nominal and adversarial accuracy.

Table 3: Adversarial robustness of models with fixed pooling operations under PGD-200 at $\epsilon = 0.05$.

| Pooling Operation | Nominal Accuracy | | | Adversarial Accuracy | | |
| --- | --- | --- | --- | --- | --- | --- |
| | PointNet | DeepSets | DSS | PointNet | DeepSets | DSS |
| MAX | 80.5% | 71.1% | 78.8% | 16.1% | 21.8% | 21.5% |
| SUM | 76.3% | 54.1% | 73.3% | **25.1%** | **24.8%** | **25.3%** |
| MEDIAN | **84.6%** | **72.7%** | **82.4%** | 7.5% | 11.0% | 9.3% |

**Analysis.** AT consists of two stages: 1) *inner maximization* to find the worst adversarial examples and 2) *outer minimization* to update model parameters. Fixed pooling operations essentially leverage

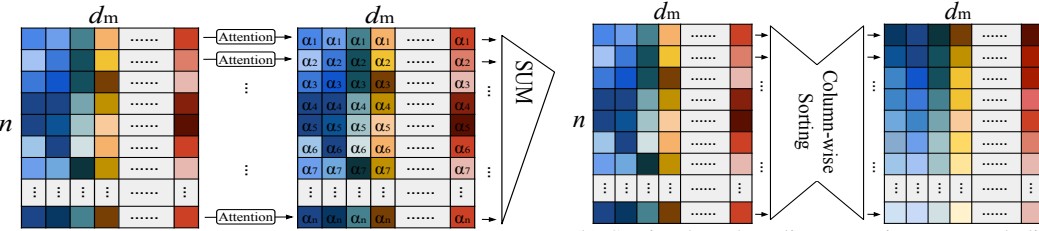

(a) Attention-based pooling operations apply self-attention to each point-level feature vector $\boldsymbol{f}_i$. The learned weight $\alpha_i$ is multiplied with each element in $\boldsymbol{f}_i$, and the aggregated feature is computed by a column-wise summation.

(b) Sorting-based pooling operations sort each dimension to re-organize the feature set into an ordered matrix $\widetilde{\boldsymbol{F}}$ to which complex operations (*e.g.,* CNN) can be applied to aggregate features.

Figure 4: Design philosophy of attention-based and sorting-based pooling operations.

a *single* statistic to represent the distribution of a feature dimension (Murray & Perronnin, 2014). Although MEDIAN pooling, as a robust statistic, intuitively should enhance the robustness, we find it actually hinders the inner maximization stage from making progress. We utilize $L^\infty$ norm-based PGD attack to maximize the xent loss of standard trained model with three fixed pooling operations. Figure 3 validates that MEDIAN pooling takes many more steps to maximize the loss. Therefore, MEDIAN pooling fails to find the worst adversarial examples in the first stage with limited steps. Though MAX and SUM pooling are able to achieve higher loss value, they encounter challenges in the second stage. MAX pooling backward propagates gradients to a *single* point at each dimension so that the rest $\frac{n-1}{n}$ features do not contribute to model learning. Since $n$ is oftentimes a large number (*e.g.,* 1024), the huge information loss and non-smoothness will fail AT (Xie et al., 2020). While SUM pooling realizes a smoother backward propagation, it lacks discriminability because by applying the same weight to each element, the resulting representations are strongly biased by the adversarial perturbations. Thus, with SUM pooling, the models cannot generalize well on clean data.

### 4.3 ADVERSARIAL TRAINING WITH PARAMETRIC POOLING OPERATIONS

Recent studies have also presented trainable **parametric pooling operations** for different tasks in set learning, *e.g.,* multiple instance learning, which are also qualified to be the symmetric function $\rho(\cdot)$ in point cloud classification models. Thus, we first group them into two classes: 1) *attention-based* and 2) *sorting-based* pooling, and further benchmark their robustness under AT in point cloud classification. It is worth noting that *none* of those parametric pooling operations are proposed to improve the adversarial robustness, and we are the first to conduct such an in-depth analysis of how they behave as the symmetric function under AT in point cloud classification.

#### 4.3.1 ATTENTION-BASED POOLING OPERATIONS

An attention module can be abstracted as mapping a query and a set of key-value pairs to an output, making the models learn and focus on the critical information (Bahdanau et al., 2014). Figure 4(a) shows the design principle of attention-based pooling, which leverages a compatibility function to learn point-level importance. The aggregated global feature is computed as a column-wise weighted sum of the local features. Two attention-based pooling operations, ATT and ATT-GATE, are first proposed for multiple instance learning (Ilse et al., 2018). Let $\mathbb{F} = \{\boldsymbol{f}_1, \boldsymbol{f}_2, \ldots, \boldsymbol{f}_n\}$ be a set of features, ATT aggregates the global feature $\boldsymbol{g}$ by:

$$\boldsymbol{g} = \sum_{i=1}^{n} a_i \cdot \boldsymbol{f}_i , \quad a_i = \frac{\exp(\boldsymbol{w}^\top \cdot \tanh(\boldsymbol{V} \cdot \boldsymbol{f}_i^\top))}{\sum_{j=1}^{n} \exp(\boldsymbol{w}^\top \cdot \tanh(\boldsymbol{V} \cdot \boldsymbol{f}_j^\top))} \tag{6}$$

where $\boldsymbol{w} \in \mathbb{R}^{L \times 1}$ and $\boldsymbol{V} \in \mathbb{R}^{L \times d_m}$ are learnable parameters. ATT-GATE improves the expressiveness of ATT by introducing another non-linear activation $\text{sigmoid}(\cdot)$ and more trainable parameters into weight learning. Furthermore, PMA (Lee et al., 2019) is proposed for general set learning, which leverages multi-head attention (Vaswani et al., 2017) for pooling. We detail the design and our implementation of ATT, ATT-GATE, and PMA in Appendix B.3, and adversarially train the backbone models with these attention-based pooling operations.

#### 4.3.2 SORTING-BASED POOLING OPERATIONS

Sorting has been recently considered in the set learning literature due to its permutation-invariant characteristic, as shown in Figure 4(b). Let $\boldsymbol{F} \in \mathbb{R}^{n \times d_m}$ be the matrix version of the feature set $\mathbb{F}$,

`FSPool` (Zhang et al., 2020) aggregates $\boldsymbol{F}$ by feature-wise sorting in a descending order:

$$\widetilde{\boldsymbol{F}}_{i,j} = sort_\downarrow(\boldsymbol{F}_{:,j})_i \; ; \quad g_j = \sum_{i=1}^{n} \boldsymbol{W}_{i,j} \cdot \widetilde{\boldsymbol{F}}_{i,j} \tag{7}$$

where $\boldsymbol{W} \in \mathbb{R}^{n \times d_m}$ are learnable parameters. Therefore, the pooled representation is column-wise weighted sum of $\widetilde{\boldsymbol{F}}$. `SoftPool` (Wang et al., 2020) re-organizes $\boldsymbol{F}$ so that its $j$-th dimension is sorted in a descending order, and picks the top $k$ point-level embeddings $\boldsymbol{F}'_j \in \mathbb{R}^{k \times d_m}$ to further form $\widetilde{\boldsymbol{F}} = [\boldsymbol{F}'_1, \boldsymbol{F}'_2, \ldots, \boldsymbol{F}'_{d_m}]$. Then, `SoftPool` applies CNN to each $\widetilde{\boldsymbol{F}}_j \to \boldsymbol{g}_j$ so that the pooled representation is $\boldsymbol{g} = [\boldsymbol{g}_1, \boldsymbol{g}_2, \ldots, \boldsymbol{g}_{d_m}]$. Implementation details of `SoftPool` are elaborated in Appendix B.3. We also adversarially train the backbone models with `FSPool` and `SoftPool`.

Table 4: Adversarial robustness of models with parametric pooling operations under PGD-200 at $\epsilon = 0.05$.

| Pooling Operation | Nominal Accuracy | | | Adversarial Accuracy | | |
|---|---|---|---|---|---|---|
| | PointNet | DeepSets | DSS | PointNet | DeepSets | DSS |
| ATT | 73.5% | 52.3% | 72.8% | 22.1% | 23.2% | 23.9% |
| ATT-GATE | 75.1% | 63.9% | 73.3% | 23.2% | 24.8% | 26.1% |
| PMA | 73.9% | 51.9% | 72.5% | 25.4% | 20.9% | 23.9% |
| FSPool | **82.8%** | 73.8% | 81.5% | 29.8% | 25.3% | 26.1% |
| SoftPool | 79.8% | 72.1% | 80.2% | 30.1% | 24.9% | 26.5% |
| DeepSym (ours) | 82.7% | **74.2%** | **81.6%** | **33.6%** | **26.9%** | **31.4%** |

### 4.3.3 EXPERIMENTAL ANALYSIS

Table 4 shows the results of AT with different parametric pooling operations. To meet the requirement of permutation-invariance, attention-based pooling is restricted to learn *point-level* importance. For example, `ATT` applies the same weight to all dimensions of a point embedding. As a result, attention barely improves the pooling operation's expressiveness as it essentially re-projects the point cloud to a single dimension (*e.g.,* $\boldsymbol{f}_i \to a_i$ in `ATT`) and differentiates them based on it, which significantly limits their discriminability. Therefore, little useful information can be learned from the attention module, explaining why they perform similarly to `SUM` pooling that applies the same weight to each point under AT, as shown in Table 4. Sorting-based pooling operations naturally maintain permutation-invariance as $sort_\downarrow(\cdot)$ re-organizes the unordered feature set $\mathbb{F}$ to an ordered feature map $\widetilde{\boldsymbol{F}}$. Thus, `FSPool` and `SoftPool` are able to further apply *feature-wise* linear transformation and CNN. The insight is that feature dimensions are mostly independent of each other, and each point expresses to a different extent in every dimension. By employing feature-wise learnable parameters, the gradients also flow smoother through sorting-based pooling operations. Table 4 validates that sorting-based pooling operations achieve much better adversarial accuracy, *e.g.,* on average, 7.3% better than `MAX` pooling while maintaining comparable nominal accuracy.

## 5 IMPROVING THE ADVERSARIAL ROBUSTNESS WITH DEEPSYM

In the above analysis, we have shed light on that sorting-based pooling operations can benefit AT in point cloud classification. We hereby explore to further improve the sorting-based pooling design inspired by existing arts. First, we notice that both `FSPool` and `SoftPool` apply $sort_\downarrow(\cdot)$ right after a ReLU function (Nair & Hinton, 2010). However, ReLU leads to some neurons being zero (Goodfellow et al., 2016), which makes $sort_\downarrow(\cdot)$ unstable. Second, recent studies have shown that AT appreciates deeper neural networks (Xie & Yuille, 2019). Nevertheless, `FSPool` only employs one linear layer to aggregate features, and `SoftPool` requires $d_m$ to be a small number. The reason is that scaling up the depth in these existing sorting-based pooling designs requires exponential growth of parameters, which will make the end-to-end learning intractable.

To address the above limitations, we propose a simple yet effective pooling operation, `DeepSym`, that embraces the benefits of sorting-based pooling and also applies deep learnable layers to the pooling process. Given a feature set after ReLU activation $\mathbb{F} \in \mathbb{R}_+{}^{n \times d_m}$, `DeepSym` first applies another linear transformation to re-map $\mathbb{F}$ into $\mathbb{R}^{n \times d_m}$ so that $\boldsymbol{f}'_i = \boldsymbol{W} \cdot \boldsymbol{f}_i{}^\top$ where $\boldsymbol{W} \in \mathbb{R}^{d_m \times d_m}$ and $\mathbb{F}' = \{\boldsymbol{f}'_1, \boldsymbol{f}'_2, \ldots, \boldsymbol{f}'_n\}$. Let $\boldsymbol{F}'$ be the matrix version of $\mathbb{F}'$, `DeepSym` also sorts $\boldsymbol{F}'$ in a descending order (Equation 7) to get $\widetilde{\boldsymbol{F}'}$. Afterwards, we apply column-wise shared MLP on $\widetilde{\boldsymbol{F}'}$:

$$g_j = \text{MLP}(\widetilde{\boldsymbol{F}'}_{:,j}) \, , \quad j = \{1, 2, \ldots, d_m\} \tag{8}$$

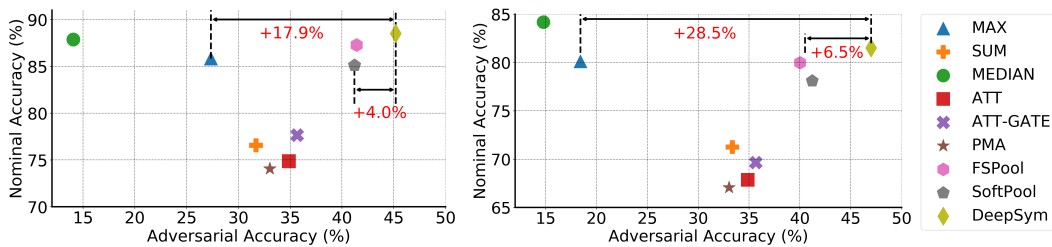

(a) PGD-1 adversarial training on ModelNet10.  (b) PGD-20 adversarial training on ModelNet40.

Figure 5: Adversarial robustness of PointNet with various pooling operations under PGD-200 at $\epsilon = 0.05$.

to learn the global feature representation $g$. Each layer of the MLP composes of a linear transformation, a batch normalization module (Ioffe & Szegedy, 2015), and a ReLU activation function. Compared to `FSPool` that applies different linear transformations to different dimensions, `DeepSym` employs a *shared* MLP to different dimensions. By doing so, `DeepSym` deepens the pooling process to be more capable of digesting the adversarial perturbations. `DeepSym` can also address the problem of `SoftPool` that is only achievable with limited $d_m$ because the MLP is shared by all the feature channels so that it can scale up to a large number of $d_m$ with little complexity increases. Moreover, `DeepSym` generalizes MAX and SUM pooling by specific weight vectors. Therefore, it can also theoretically achieve universality with $d_m \geq n$ (Wagstaff et al., 2019) while being more expressive in its representation and smoother in gradients propagation. To deal with the variable-size point clouds, `DeepSym` adopts column-wise linear interpolation in $\widetilde{F'}$ to form a continuous feature map and then re-samples it to be compatible with the trained MLP (Jaderberg et al., 2015). Last but not least, `DeepSym` is by design flexible with the number of pooled features from each dimension. In the paper, we only allow DeepSym to output a single feature for a fair comparison with others. However, it is hard for other pooling operations to have this ability. For example, it requires a linear complexity increase for `FSPool` to enable this capability.

## 5.1 EVALUATIONS

We implement a 5-layer `DeepSym` with $[512, 128, 32, 8, 1]$ hidden neurons on three backbone networks and adversarially train them on ModelNet40 the same way introduced in Section 4.1. Table 4 shows that almost all models with `DeepSym` reach the best results in both nominal and adversarial accuracy, outperforming the default architecture by 10.8%, on average. Taking PointNet as an example, `DeepSym` (33.6%) improves the adversarial accuracy by 17.5% ($\sim 2.1\times$) compared to the original MAX pooling architecture. Besides, `DeepSym` also achieves a 3.5% improvement in adversarial accuracy compared to `FSPool` and `SoftPool`. Overall, we demonstrate that `DeepSym` can benefit AT significantly in point cloud classification.

We further leverage various white- and black-box adversarial attacks to cross validate the robustness improvements of `DeepSym` on PointNet. Specifically, we exploit well-known FGSM (Szegedy et al., 2013), BIM (Kurakin et al., 2016), and MIM (Dong et al., 2018) as the white-box attack methods. We set the adversarial budget $\epsilon = 0.05$, and leverage 200 steps for the iterative attacks, as well. For the black-box attacks, we choose two score-based methods: SPSA (Uesato et al., 2018) and NES (Ilyas et al., 2018), and a decision-based evolution attack (Dong et al., 2020b). We still select $\epsilon = 0.05$ and allow 2000 queries to find each adversarial example. The detailed setups are elaborated in Appendix C.1. As shown in Table 5, PointNet with `DeepSym` consistently achieves the best adversarial accuracy under white-box attacks, except for FGSM. The reason is that FGSM is a single-step method that has limited ability to find adversarial examples. Besides, we find the black-box attacks are not as effective as the white-box attacks, which also demonstrate that adversarial training with `DeepSym` is able to improve the robustness of point cloud classification without gradient obfuscation (Carlini et al., 2019).

Since `DeepSym` brings deep trainable layers into the original backbones, it is necessary to report its overhead. We leverage TensorFlow (Abadi et al., 2016) and NVIDIA profilers to measure the inference time, the number of trainable parameters, and GPU memory usage on PointNet. Specifically, the inference time is averaged from 2468 objects in the validation set, and the GPU memory is measured on an RTX 2080 with batch size = 8. As shown in Table 6, `DeepSym` indeed introduces more computation overhead by leveraging the shared MLP. However, we believe the overhead is relatively small and acceptable, compared to its massive improvements on the adversarial robustness. To fur-

Table 5: Adversarial robustness of PointNet with different pooling operations under attacks at $\epsilon = 0.05$.

| Pooling Operation | White-box Attack | | | Black-box Attack | | |
|---|---|---|---|---|---|---|
| | FGSM | BIM | MIM | SPSA | NES | Evolution |
| MAX | 72.8% | 24.3% | 23.5% | 69.2% | 67.1% | 53.4% |
| MEDIAN | **77.6%** | 23.3% | 14.5% | 71.1% | 65.2% | 57.8% |
| SUM | 44.4% | 33.5% | 37.5% | 65.3% | 62.3% | 52.7% |
| ATT | 43.1% | 33.1% | 35.0% | 68.1% | 64.8% | 55.9% |
| ATT-GATE | 43.9% | 34.2% | 33.9% | 70.2% | 65.9% | 55.8% |
| PMA | 47.2% | 31.9% | 30.1% | 67.2% | 64.1% | 53.4% |
| FSPool | 61.3% | 45.4% | 48.0% | **72.8%** | 71.9% | 69.9% |
| SoftPool | 62.1% | 47.6% | 45.1% | 69.2% | 68.5% | 70.0% |
| DeepSym (ours) | 61.4% | **52.5%** | **55.4%** | 72.4% | **72.1%** | **73.1%** |

ther have a lateral comparison, point cloud classification backbones are much more light-weight than image classification models. For example, ResNet-50 (He et al., 2016) and VGG-16 (Simonyan & Zisserman, 2014) have 23 and 138 million trainable parameters, respectively, and take much longer time to do the inference. The reason that models with SoftPool and PMA have fewer trainable parameters is that they limit the number of dimensions in the global feature by design.

Table 6: Overhead measurement of PointNet with different pooling operations.

| | Inference Time (ms) | # Trainable Parameters | GPU Memory (MB) |
|---|---|---|---|
| MAX | 2.21 | 815,336 | 989 |
| MEDIAN | 2.44 | 815,336 | 989 |
| SUM | 2.23 | 815,336 | 989 |
| ATT | 2.71 | 1,340,649 | 1980 |
| ATT-GATE | 3.07 | 1,865,962 | 2013 |
| PMA | **2.10** | 652,136 | 981 |
| FSPool | 2.89 | 1,863,912 | 1005 |
| SoftPool | 2.85 | **355,328** | **725** |
| DeepSym (ours) | 3.10 | 1,411,563 | 2013 |

## 5.2 EXPLORING THE LIMITS OF DEEPSYM

There is a trade-off between the training cost and adversarial robustness in AT. Increasing the number of PGD attack steps can create harder adversarial examples (Madry et al., 2018), which could further improve the model's robustness. However, the training time also increases linearly with the number of attack iterations increasing. Due to PointNet's broad adoption (Guo et al., 2020), we here analyze how it performs under various AT settings. Specifically, we exploit the most efficient AT with PGD-1 on ModelNet10 (Wu et al., 2015), a dataset consisting of 10 categories with 4899 objects, and a relatively expensive AT with PGD-20 on ModelNet40 to demonstrate the effectiveness of DeepSym. Other training setups are identical to Section 4.1.

Figure 5 shows the results of the robustness of adversarially trained PointNet with various pooling operations under PGD-200. We demonstrate that PointNet with DeepSym still reaches the best adversarial accuracy of 45.2% under AT with PGD-1 on ModelNet10, which outperforms the original MAX pooling by 17.9% ($\sim 1.7\times$) and SoftPool by 4.0%. Surprisingly, PointNet with DeepSym also achieves the best nominal accuracy of 88.5%. Moreover, DeepSym further advances itself under AT with PGD-20 on ModelNet40. Figure 5(b) shows that PointNet with DeepSym reaches the best 47.0% adversarial accuracy, which are 28.5% ($\sim 2.6\times$) and 6.5% improvements compared to MAX pooling and SoftPool, respectively while maintaining competent nominal accuracy. We also report detailed evaluations using different PGD attack steps and budgets $\epsilon$ in Appendix C.1.

## 6 CONCLUSION

In this work, we perform the *first* rigorous study on the adversarial robustness of point cloud classification. We design adaptive attacks and demonstrate that state-of-the-art defenses against adversarial point clouds cannot provide real robustness. Furthermore, we conduct a thorough analysis of how the required symmetric function affects the AT performance of point cloud classification models. We are the first to identify that the fixed pooling generally weakens the models' robustness under AT, and on the other hand, *sorting-based* parametric pooling benefits AT well. Lastly, we propose DeepSym that further architecturally advances the adversarial accuracy of PointNet to 47.0% under AT, outperforming the original design and a strong baseline by 28.5% ($\sim 2.6\times$) and 6.5%.

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

## A   ADAPTIVE ATTACK EXPERIMENTAL SETUP AND VISUALIZATIONS

### A.1   EXPERIMENTAL SETUPS

Since **DUP-Net** is open-sourced, we target the publicly released PointNet and PU-Net models. For the $L^2$ norm-based C&W attack, we set the loss function as:

$$\mathcal{L} = (\max_{i \neq t'}(\mathcal{Z}(\boldsymbol{X}')_i) - \mathcal{Z}(\boldsymbol{X}')_{t'})^+ + \lambda \cdot ||\boldsymbol{X} - \boldsymbol{X}'||_2 \qquad (9)$$

where $\boldsymbol{X} \in \mathbb{R}^{n \times 3}$ is the matrix version of point cloud $\mathbb{X}$, $\boldsymbol{X}'$ is the optimized adversarial example, $\mathcal{Z}(\boldsymbol{X})_i$ is the $i$-th element of the output logits, and $t'$ is the target class. We leverage 10-step binary search to find the appropriate hyper-parameter $\lambda$ from $[10, 80]$. As suggested by Xiang et al. (2019), we choose 10 distinct classes and pick 25 objects in each class from the ModelNet40 validation set for evaluation. The step size of the adversarial optimization is 0.01 and we allow at most 500 iterations of optimization in each binary search to find the adversarial examples.

For the $L^\infty$ norm-based PGD attack, we adopt the formulation in Madry et al. (2018):

$$\boldsymbol{X}_{t+1} = \Pi_{\boldsymbol{X}+\mathcal{S}}(\boldsymbol{X}_t + \alpha \cdot \text{sign}(\nabla_{\boldsymbol{X}_t}\mathcal{L}(\boldsymbol{X}_t, \boldsymbol{\theta}, \boldsymbol{y}))) \qquad (10)$$

where $\boldsymbol{X}_t$ is the adversarial example in the $t$-th attack iteration, $\Pi$ is the projection function to project the adversarial example to the pre-defined perturbation space $\mathcal{S}$, which is the $L^\infty$ norm ball in our setup, and $\alpha$ is the step size. We select the boundary of allowed perturbations $\epsilon = \{0.01, 0.025, 0.05, 0.075\}$ out of the point cloud data range $[-1, 1]$. Since point cloud data is continuous, we set the step size $\alpha = \frac{\epsilon}{10}$.

For **GvG-PointNet++**, we train it based on the single scale grouping (SSG)-PointNet++ backbone. The backbone network has three PointNet set abstraction module to hierarchically aggregate local features, and we enable gather vectors in the last module, which contains 128 local features (*i.e.,* $n' = 128$ in Section 3.2) with 256 dimensions. To learn the gather vectors, we apply three fully connected layers with 640, 640, and 3 hidden neurons respectively, as suggested by Dong et al. (2020a). Since the data from ModelNet40 is normalized to [-1,1], the global object center is $\boldsymbol{c}_g = [0, 0, 0]$.

For the $L^\infty$ norm-based PGD attack, we leverage the same setup as the attack on DUP-Net. For the $L^2$ norm-based PGD attack, we follow the settings in Dong et al. (2020a) to set the $L^2$ norm threshold $\epsilon = \delta\sqrt{n \times d_{in}}$, where $\delta$ is selected in $\{0.08, 0.16, 0.32\}$, $n$ is the number of points, and $d_{in}$ is the dimension of input point cloud (*i.e.,* 3). The attack iteration is set to 50, and the step size $\alpha = \frac{\epsilon}{50}$.

### A.2   VISUALIZATIONS

We visualize some adversarial examples generated by adaptive attacks on PU-Net and DUP-Net in Figure 6 and Figure 7. It is expected that adversarial examples targeting DUP-Net are noisier than the ones targeting PU-Net as the former needs to break the denoiser layer. However, as mentioned in Section 3.1, they are barely distinguishable from human perception. We also visualize some adversarial examples generated by untargeted adaptive PGD attacks on GvG-PointNet++ in Figure 8 with different perturbation budgets $\epsilon$.

## B   ADVERSARIAL TRAINING SETUP

### B.1   PGD ATTACK IN ADVERSARIAL TRAINING

We also follow the formulation in Equation 18 to find the worst adversarial examples. Specifically, we empirically select $\epsilon = 0.05$ into the training recipe as there is no quantitative study on how much humans can bear the point cloud perturbations. Figure 8 shows that adversarial examples with $\epsilon = 0.05$ are still recognizable by human perception. Moreover, because point cloud data is continuous, we set the step size of PGD attacks as:

$$\alpha = \begin{cases} \dfrac{\epsilon}{\text{step}}, & \text{step} < 10 \\ \dfrac{\epsilon}{10}, & \text{step} \geq 10 \end{cases} \qquad (11)$$

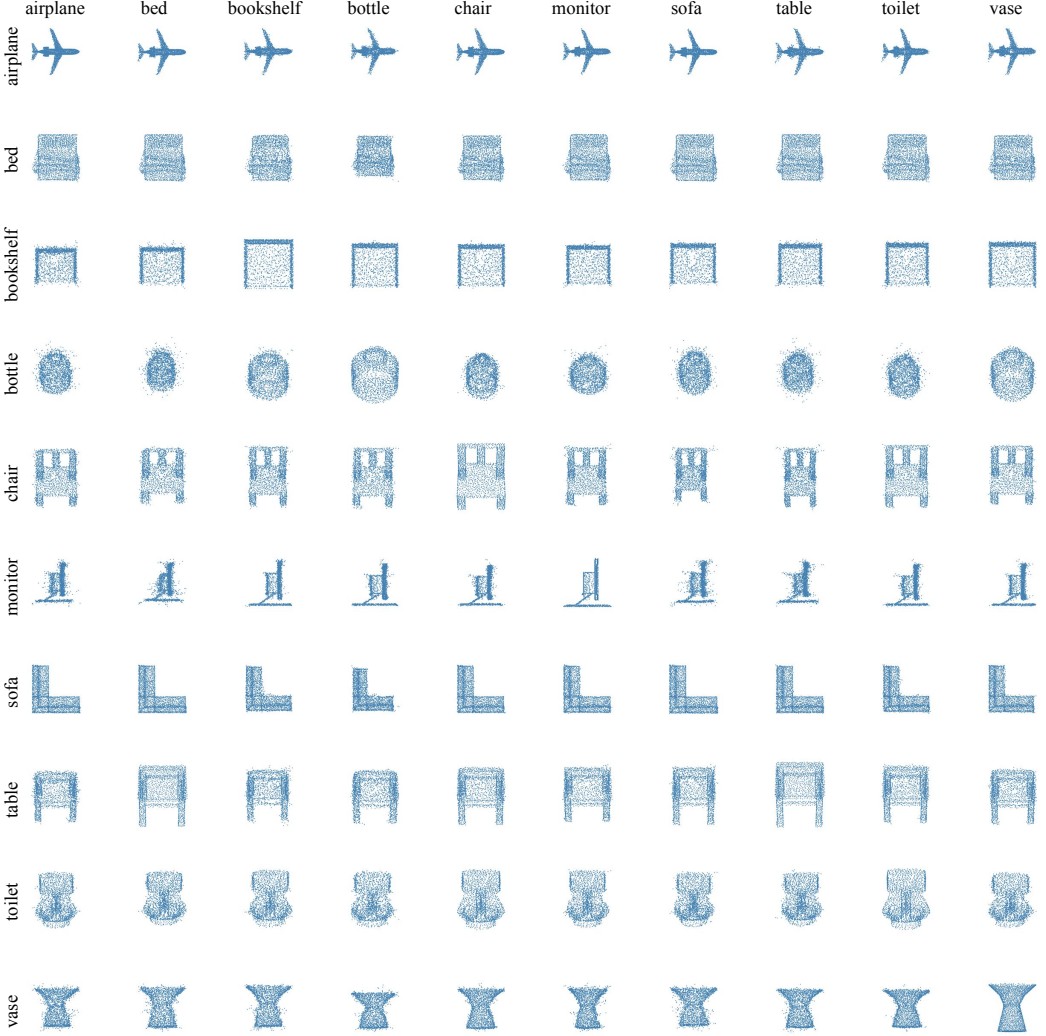

Figure 6: Visualizations of adversarial examples (2048 points) generated by $L^2$ norm-based C&W adaptive attacks on PU-Net.

in both training and evaluation phases to make sure PGD attacks reach the allowed maximum perturbations.

## B.2  POINT CLOUD CLASSIFICATION MODEL ARCHITECTURE DETAILS

PointNet, DeepSets, and DSS are the fundamental architectures in point cloud classification. Other models, such as PointNet++ and DGCNN, are built upon PointNet and DeepSets. Moreover, complex models oftentimes apply non-differentiable layers like $knn(\cdot)$ into end-to-end learning, which will make the adversarial training ineffective. In this work, we aim at exploring how the symmetric (permutation-invariant) function can benefit adversarial training. To this end, we choose PointNet, DeepSets, and DSS as the backbone networks. For the ModelNet40 dataset, we follow the default setting to split into 9,843 objects for training and 2,468 objects for validation (Wu et al., 2015). We randomly sample 1024 points from each object to form its point cloud, if not otherwise stated.

**PointNet.** We leverage the default architecture in PointNet codebase[3] and exclude the transformation nets (*i.e.,* T-Net) and dropout layers for simplicity and reproducibility. PointNet leverages shared fully connected (FC) layers as the permutation-equivariant layer $\phi_l : \text{FC}_l(\boldsymbol{F}_{l:,i}) \rightarrow \boldsymbol{F}_{l+1:,i}$ and MAX pooling as the symmetric function $\rho(\cdot)$.

---

[3]https://github.com/charlesq34/pointnet

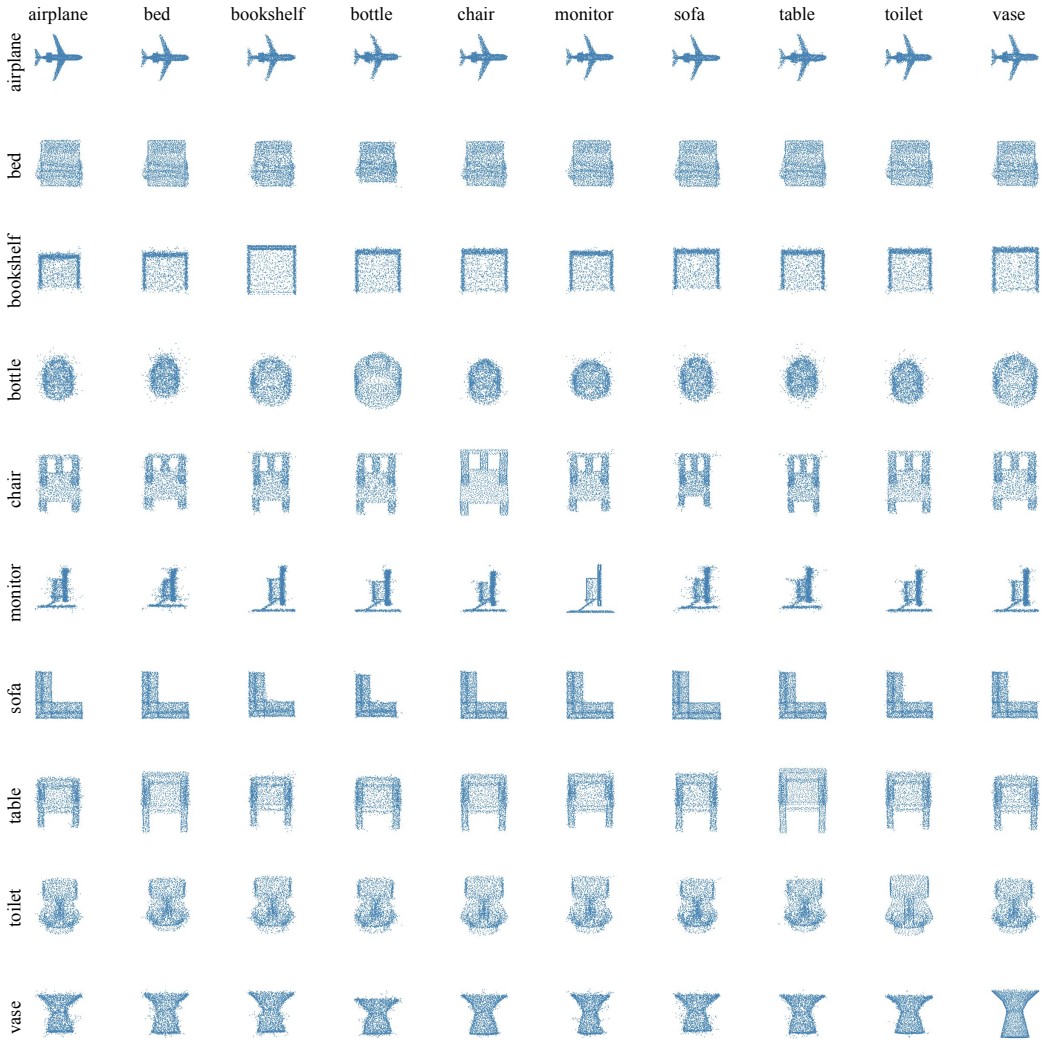

Figure 7: Visualizations of adversarial examples (2048 points) generated by $L^2$ norm-based C&W adaptive attacks on DUP-Net.

**DeepSets.** We leverage the default architecture in DeepSets codebase[4]. Different from PointNet, DeepSets first applies a symmetric function to each feature map and aggregate it with the original feature map. Afterwards, DeepSets also leverages FC layers to further process the features: $\phi_l : \text{FC}_l(\boldsymbol{F}_{l:,i} - \zeta(\boldsymbol{F}_l)) \rightarrow \boldsymbol{F}_{l+1:,i}$, where $\zeta(\cdot)$ is column-wise MAX pooling in the original implementation. Similarly, MAX pooling is still used as $\rho(\cdot)$ in DeepSets.

**DSS.** DSS generalizes DeepSets architecture and applies another FC layer to $\zeta(\boldsymbol{F}_l)$ in DeepSets so that $\phi_l : \text{FC}_{l1}(\boldsymbol{F}_{l:,i}) + \text{FC}_{l2}(\zeta(\boldsymbol{F}_l)) \rightarrow \boldsymbol{F}_{l+1:,i}$. Different from other two achitectures, DSS utilizes SUM pooling as $\rho(\cdot)$. Since there is no available codebase at the time of writing, we implement DSS by ourselves.

We visualize the differences of $\phi(\cdot)$ in Figure 9, and summarize the layer information in Table 7.

### B.3 PARAMETRIC POOLING DESIGN AND IMPLEMENTATION

We have introduced ATT in Section 4.3.1. In our implementation, we choose $L = 512$ so that $\boldsymbol{V} \in \mathbb{R}^{512 \times 1024}$ to train the backbone models.

---

[4]https://github.com/manzilzaheer/DeepSets

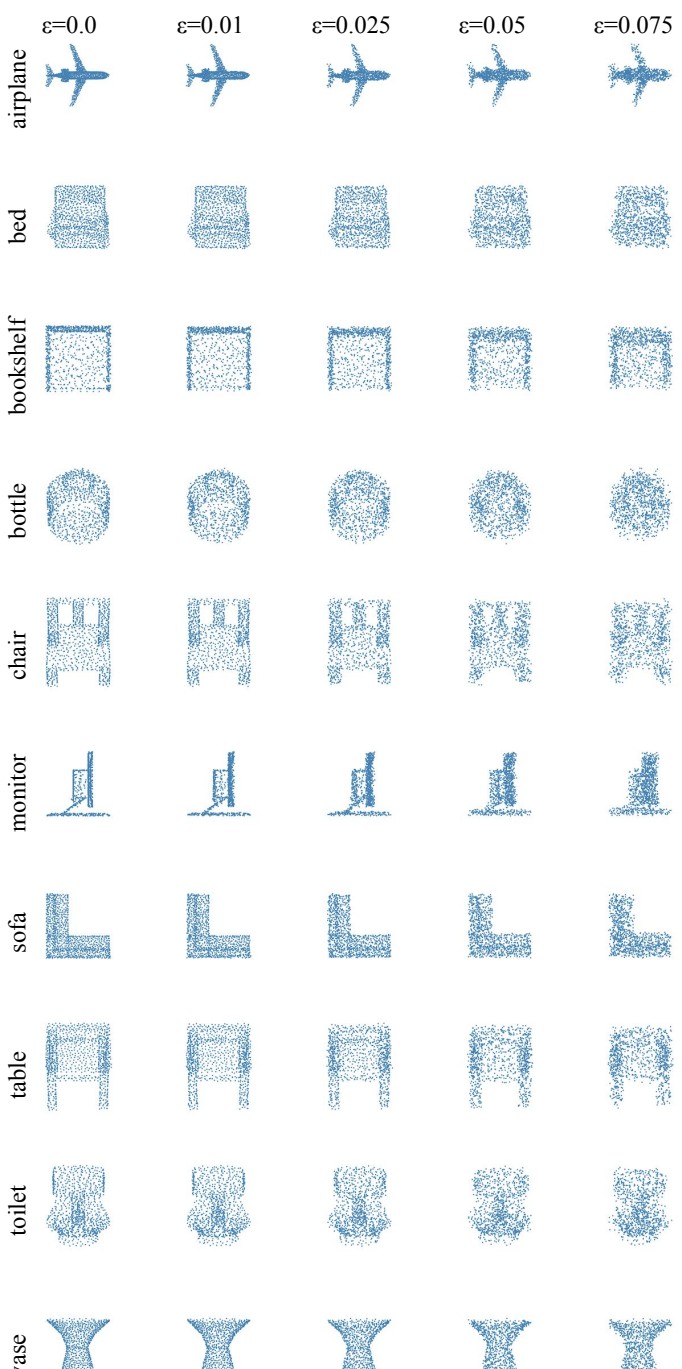

Figure 8: Visualizations of adversarial examples (1024 points) generated by $L^\infty$ norm-based PGD adaptive attacks on GvG-PointNet++.

`ATT-GATE` is a variant of `ATT` with more learnable parameters:

$$\boldsymbol{g} = \sum_{i=1}^{n} a_i \cdot \boldsymbol{f}_i \; , \quad a_i = \frac{\exp(\boldsymbol{w}^\top \cdot (\tanh(\boldsymbol{V} \cdot \boldsymbol{f}_i^\top) \odot \text{sigm}(\boldsymbol{U} \cdot \boldsymbol{f}_i^\top)))}{\sum_{j=1}^{n} \exp(\boldsymbol{w}^\top \cdot (\tanh(\boldsymbol{V} \cdot \boldsymbol{f}_j^\top) \odot \text{sigm}(\boldsymbol{U} \cdot \boldsymbol{f}_j^\top)))} \tag{12}$$

where $\boldsymbol{U}, \boldsymbol{V} \in \mathbb{R}^{L \times M}$, $\text{sigm}(\cdot)$ is the sigmoid activation function, and $\odot$ is an element-wise multiplication. We also choose $L = 512$ in `ATT-GATE` to train the backbone models.

Table 7: Layer information of PointNet, DeepSets, and DSS. BN represents a batch normalization layer.

| PointNet | DeepSets | DSS |
|---|---|---|
| $\phi_1 : n \times 3 \to n \times 64$ | $\phi_1 : n \times 3 \to n \times 256$ | $\phi_1 : n \times 3 \to n \times 64$ |
| BN + ReLU | BN + ELU | BN + ReLU |
| $\phi_2 : n \times 64 \to n \times 64$ | $\phi_2 : n \times 256 \to n \times 256$ | $\phi_2 : n \times 64 \to n \times 256$ |
| BN + ReLU | BN + ELU | BN + ReLU |
| $\phi_3 : n \times 64 \to n \times 128$ | $\rho : n \times 256 \to 256$ | $\phi_3 : n \times 256 \to n \times 256$ |
| BN + ReLU | $\sigma_1 : 256 \to 256$ | BN + ReLU |
| $\phi_4 : n \times 128 \to n \times 1024$ | BN + Tanh | $\rho : n \times 256 \to 256$ |
| BN + ReLU | $\sigma_2 : 256 \to 40$ | $\sigma_1 : 256 \to 256$ |
| $\rho : n \times 1024 \to 1024$ | | BN + ReLU |
| $\sigma_1 : 1024 \to 512$ | | $\sigma_2 : 256 \to 40$ |
| BN + ReLU | | |
| $\sigma_2 : 512 \to 256$ | | |
| BN + ReLU | | |
| $\sigma_3 : 256 \to 40$ | | |

(a) $\phi(\cdot)$ in PointNet.  (b) $\phi(\cdot)$ in DeepSets.

(c) $\phi(\cdot)$ in DSS.  (d) The aggregated feature in (b) and (c).

Figure 9: Different architectures of $\phi(\cdot)$ in PointNet, DeepSets, and DSS.

PMA (Lee et al., 2019) adopts multi-head attention into pooling on a learnable set of $k$ seed vectors $\boldsymbol{S} \in \mathbb{R}^{k \times d_m}$ Let $\boldsymbol{F} \in \mathbb{R}^{n \times d_m}$ be the matrix version of the set of features.

$$\text{PMA}_k(\boldsymbol{F}) = \text{MAB}(\boldsymbol{S}, \text{FC}(\boldsymbol{F})) \tag{13}$$

$$\text{MAB}(\boldsymbol{X}, \boldsymbol{Y}) = \boldsymbol{H} + \text{FC}(\boldsymbol{H}) \tag{14}$$

$$\text{where} \quad \boldsymbol{H} = \boldsymbol{X} + \text{Multihead}(\boldsymbol{X}, \boldsymbol{Y}, \boldsymbol{Y}; \boldsymbol{w}) \tag{15}$$

where $\text{FC}(\cdot)$ is the fully connected layer and $\text{Multihead}(\cdot)$ is the multi-head attention module (Vaswani et al., 2017). We follow the implementation in the released codebase[5] to choose $k = 1$, the number of head = 4, and the hidden neurons in $\text{FC}(\cdot) = 128$ to train the backbone models.

Since SoftPool (Wang et al., 2020) sorts the feature set in each dimension, it requires the number of dimensions $d_m$ to be relatively small. We follow the description in their paper to choose $d_m = 8$ and $k = 32$ so that each $\boldsymbol{F}_j' \in \mathbb{R}^{32 \times 8}$. We apply one convolutional layer to aggregate each $\boldsymbol{F}_j'$ into $g_j \in \mathbb{R}^{1 \times 32}$ so that the final $\boldsymbol{g} \in \mathbb{R}^{1 \times 256}$. Therefore, for all backbone networks with SoftPool, we apply the last equivariant layer as $\phi : n \times d_{m-1} \to n \times 8$ and $\rho : n \times 8 \to 256$.

---

[5] https://github.com/juho-lee/set_transformer

# C  DEEPSYM ABLATIONS

It is worth noting that `DeepSym` does not require the final layer to have only one neuron. However, to have a fair comparison with other pooling operations that aggregate into one feature from each dimension, our implementation of `DeepSym` also aggregates into one feature from each dimension.

## C.1  EVALUATION DETAILS

We also perform extensive evaluations using different PGD attack steps and budgets $\epsilon$ on PGD-20 trained PointNet. Figure 10 shows that PointNet with `DeepSym` consistently achieves the best adversarial accuracy. We also validate `MEDIAN` pooling indeed hinders the gradient backward propagation. The adversarial accuracy of PointNet with `MEDIAN` pooling consistently drops even after PGD-1000. However, the adversarial accuracy of PointNet with other pooling operations usually converges after PGD-200. Figure 11 shows that `DeepSym` also outperforms other pooling operations under different adversarial budgets $\epsilon$.

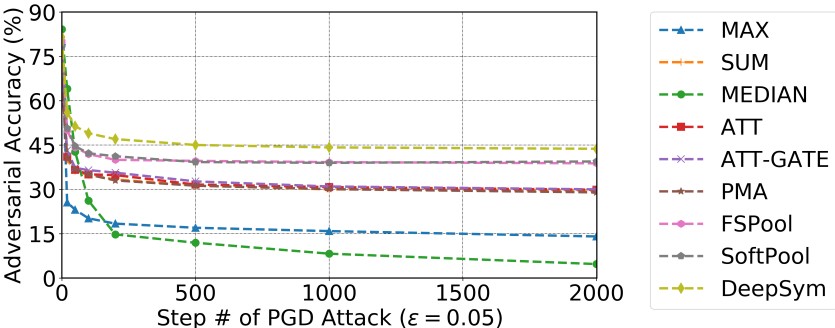

Figure 10: Adversarial accuracy of PGD-20 trained PointNet with different pooling operations. We leverage the PGD attack with different steps to evaluate the model's robustness.

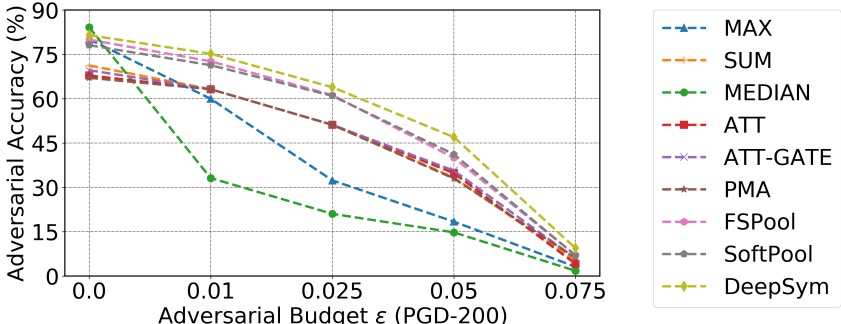

Figure 11: Adversarial accuracy of PGD-20 trained PointNet with different pooling operations. We leverage the PGD attack with different budgets to evaluate the model's robustness.

We leverage the default setup in FGSM, BIM, and MIM in our evaluation. FGSM is a single-step attack method, which can be represented as:

$$X_{adv} = X + \epsilon \cdot \text{sign}(\nabla_X \mathcal{L}(X, \theta, y)) \tag{16}$$

The BIM attack is similar to PGD attacks described in Appendix A.1. The differences are 1) the attack starts from the original point cloud $X$ and 2) the step size $\alpha = \epsilon/T$, where $T$ is the number of attack steps. The MIM attack introduces momentum terms into the adversarial optimization:

$$g_{t+1} = \mu \cdot g_t + \frac{\nabla_{X_t} \mathcal{L}(X_t, \theta, y)}{||\nabla_{X_t} \mathcal{L}(X_t, \theta, y))||_1} \tag{17}$$

$$X_{t+1} = X_t + \alpha \cdot \text{sign}(g_{t+1}) \tag{18}$$

Similar to BIM, the attack starts from the original point cloud $X$ and the step size $\alpha = \epsilon/T$. We set $\mu = 1$ following the original setup (Dong et al., 2018).

Due to the computational resource constraints, we set the sample size = 32 and allow 2000 quires to find each adversarial example in the score-based black-box attack (Uesato et al., 2018; Ilyas et al., 2018). For the evolution attack, we use the default loss $\mathcal{L}$ as the fitness score, and initialize 32 sets of perturbations from a Gaussian distribution $\mathcal{N}(0, 1)$. 4 sets of perturbations with top fitness scores will remain for the next iteration, while others will be discarded. We also allow 2000 generations of evolution to find the adversarial example.

## C.2  EVALUATION ON SCANOBJECTNN

We also evaluate the adversarial robustness of different pooling operations on a new point cloud dataset, ScanObjectNN (Uy et al., 2019), which contains 2902 objects belonging to 15 categories. We leverage the same adversarial training setup as ModelNet10 (*i.e.,* PGD-1). Table 8 shows the results. We find that PointNet with `DeepSym` still achieves the best adversarial robustness. Since the point clouds from ScanObjectNN are collected from real-world scenes, which suffers from occlusion and imperfection, both nominal and adversarial accuracy drops compared to the results ModelNet40. We find that even some clean point clouds cannot be correctly recognized by human perception. Therefore, the performance degradation is also expected and we believe the results are not as representative as ones on ModelNet40.

Table 8: Adversarial robustness of PointNet with different pooling operations under PGD-200 at $\epsilon = 0.05$.

| Pooling Operation | Nominal Accuracy | Adversarial Accuracy |
|:---:|:---:|:---:|
| MAX | 75.2% | 16.8% |
| MEDIAN | 68.4% | 8.2% |
| SUM | 63.5% | 18.3% |
| ATT | 62.7% | 17.9% |
| ATT-GATE | 59.8% | 17.1% |
| PMA | 61.2% | 16.2% |
| FSPool | **76.8%** | 20.1% |
| SoftPool | 73.2% | 17.2% |
| DeepSym (ours) | 76.7% | **22.8%** |

## C.3  T-SNE VISUALIZATIONS

We visualize the global feature embeddings of adversarially trained PointNet under PGD-20 with different pooling operations in Figure 12 and their logits in Figure 13. Since it is hard to pick 40 distinct colors, though we put all data from 40 classes into the T-SNE process, we only choose 10 categories from ModelNet40 to realize the visualizations.

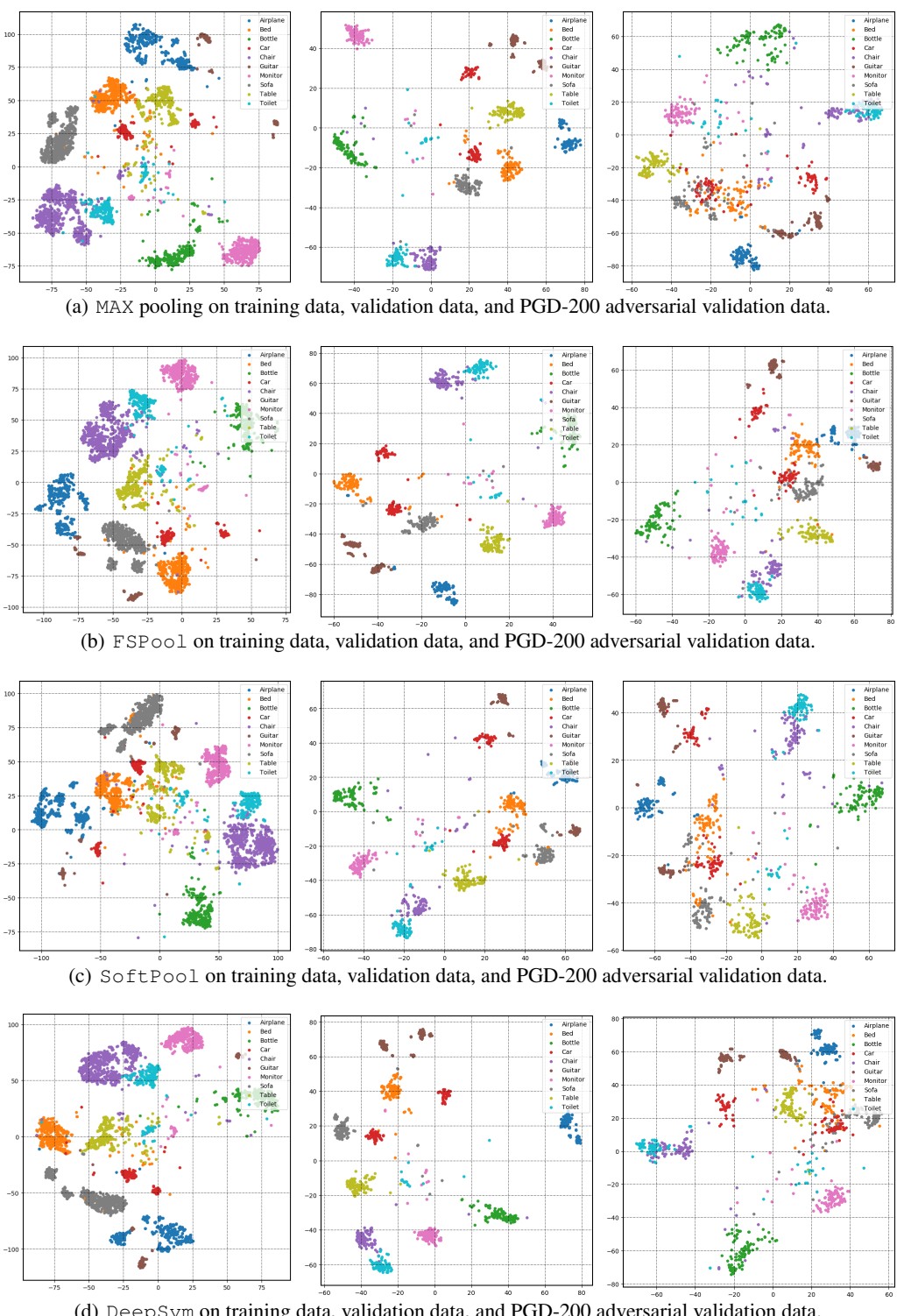

Figure 12: T-SNE visualizations of PointNet feature embeddings with MAX, FSPool, SoftPool, and DeepSym pooling operations. Three columns correspond to training data, validation data, and PGD-200 adversarial validation data, from left to right.

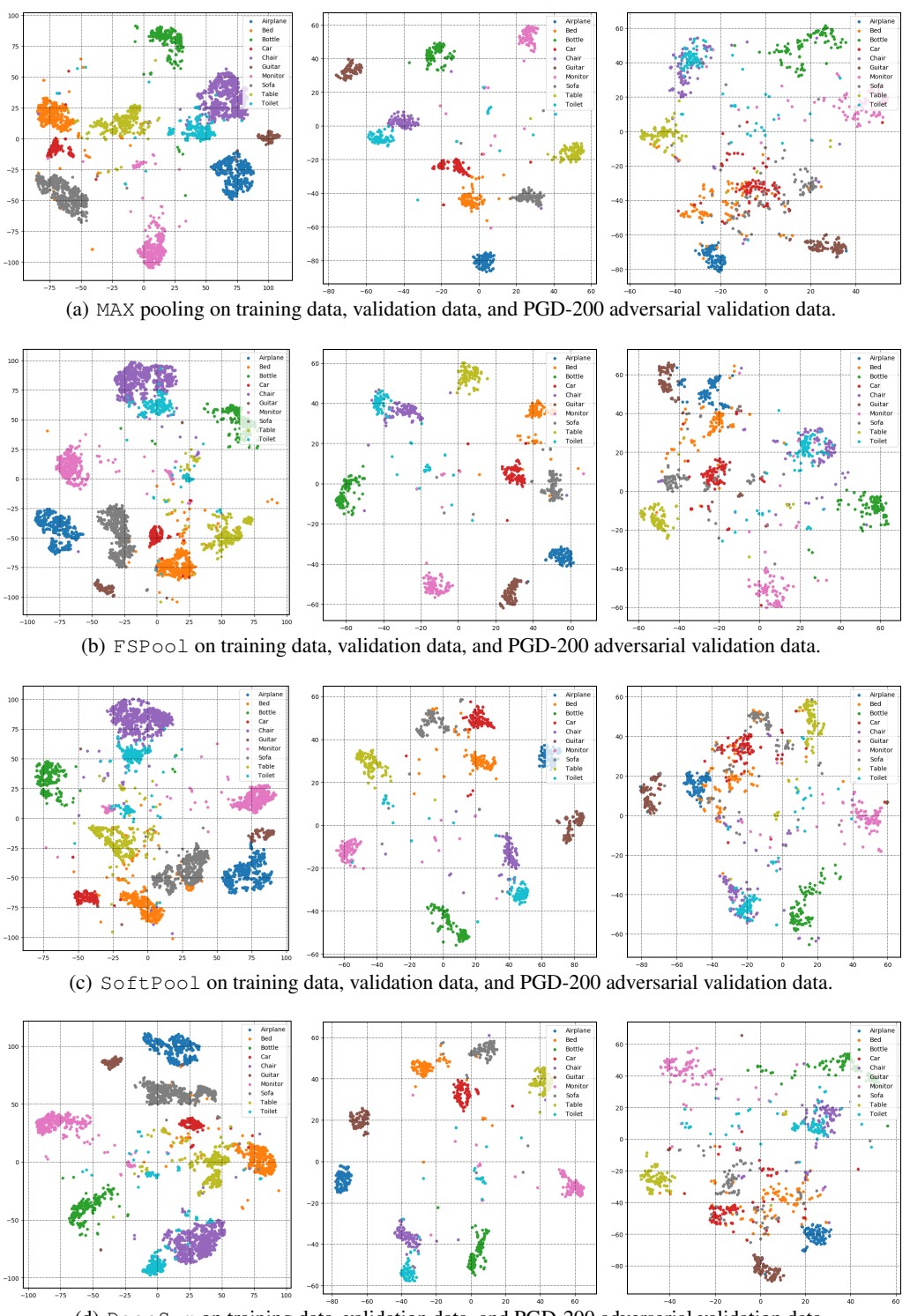

(a) MAX pooling on training data, validation data, and PGD-200 adversarial validation data.

(b) FSPool on training data, validation data, and PGD-200 adversarial validation data.

(c) SoftPool on training data, validation data, and PGD-200 adversarial validation data.

(d) DeepSym on training data, validation data, and PGD-200 adversarial validation data.

Figure 13: T-SNE visualizations of PointNet logits with MAX, FSPool, SoftPool, and DeepSym pooling operations. Three columns correspond to training data, validation data, and PGD-200 adversarial validation data, from left to right.

