# OpenReview forum: "On The Adversarial Robustness of 3D Point Cloud Classification"
_ICLR.cc/2021/Conference — Reject_

### Official Review · AnonReviewer3 · 2020-10-28
**The paper proves that state-of-the-art defenses against 3D adversarial point clouds are still vulnerable to adaptive attacks, and proposes a deep symmetric pooling operatio, DeepSym, to improve the model’s robustuness.**

**Rating:** 5
**Confidence:** 2

**Review:**

Pros:
1. The paper performs the first security analysis of defenses and designs adaptive attacks, which demonstrate current defense designs are vulnerable.
2. The paper proves that sorting-based parametric pooling operations can improve the model’s robustness.
3. Based on the existing problems, a deep symmetric pooling operation, DeepSym, is proposed.

Cons:
1. DeepSym seems to be simple and slightly lacking in novelty.
2. Unlike existing methods, DeepSym applies deeper neural networks. Therefore, the computational cost (both time and space consuming) of the DeepSym should be discussed in the paper.

---

> ### Author Response · Authors · 2020-11-14
> **Response to Reviewer3's questions (Part 1)**
>
> We thank the reviewer for the constructive comments. Below we respond to each question of the reviewer:
>
> [**Novelty**] We agree that DeepSym is a simple architecture, but we instead view its simplicity as an advantage. We also believe that our paper has novelty, as echoed in the response to Reviewer1.
>
> First, although parametric pooling operations have been studied in the literature, no existing works study their applications on the adversarial robustness of point cloud classification, which has important security implications in application domains such as autonomous driving. Previous work FSPool targets on point cloud autoencoders, and PMA was proposed for set structure learning. Therefore, one of our contributions is to unveil and benchmark the robustness of different pooling operations as the symmetric function in point cloud learning.  Specifically, one part of the novelty of our paper comes from the identification that the symmetric function significantly affects the robustness of point cloud classification. In summary, the contribution of this paper is three-fold. The adaptive attacks on state-of-the-art defenses first motivate us to study the adversarial training performance on point cloud classification. The success of DeepSym is based on the insights from the failure of fixed pooling operations and the comprehensive benchmarking of parametric pooling operations. Thus, DeepSym is actually built upon multiple valuable findings from the first part of our paper, and we believe the integrated contributions are strong enough to make our paper stand out.
>
> Second, we believe DeepSym itself has novelty. The key difference between DeepSym and others is the usage of a shared MLP. The original FSPool applies $n \times d_m$ learnable weights to the sorted feature map, which is a large number of parameters to train, and the linear transformation also lacks expressiveness. This design further limits FSPool to go deeper since the number of parameters will exponentially increase. The design of SoftPool is very complicated so that it requires the $d_m$ to be a small number to make the training tractable. Thus, SoftPool cannot achieve universality, which requires $d_m >= n$ \[1\]. In contrast, DeepSym has fairly good expressiveness since the deep networks are shown to have much more capacity, which also explains why DeepSym achieves good performance in both nominal and adversarial accuracy. Since the MLP is shared by different feature dimensions, the complexity increase of DeepSym is also well balanced. We will quantitatively show it in our second response to address the overhead concerns. Besides, DeepSym can also realize universality at the same time since it accepts $d_m >= n$. Moreover, DeepSym is by design flexible with the number of pooled features from each dimension. In the paper, we only allow DeepSym to output a single feature for a fair comparison with others. However, it is hard for other pooling operations to have this ability. For example, it requires a linear complexity increase for FSPool to enable this ability. However, DeepSym can achieve it with barely any complexity increases.
>
> Third, we also believe that the performance improvement should be a more important factor than the design complexity. The robustness enhancement by DeepSym is indeed significant. In the area of adversarial machine learning, various ideas that claimed to dramatically improve the adversarial robustness have been eventually demonstrated to be **gradient hiding** based methods\[2\], like DUP-Net and GvG-PointNet++. More papers with new ideas are broken by simple adaptive attacks very recently \[3\]. Such “novel” ideas actually cannot provide real adversarial robustness. As suggested in “On Evaluating Adversarial Robustness” by Carlini et al. \[4\], it is important for one paper to conduct correct evaluations rather than novel ideas without justification. Adversarial training-based methods are widely believed as the most effective way to mitigate the adversarial attacks. Thus, we believe that as the first study on the **real** robustness of point cloud classification, an absolute 28.5% ($2.6 \times$) and 6.5% improvement by DeepSym compared to Max and FSPool respectively should be considered significant.
>
> [1] Edward Wagstaff, Fabian B Fuchs, Martin Engelcke, Ingmar Posner, and Michael Osborne. On the limitations of representing functions on sets. arXiv preprint arXiv:1901.09006, 2019.
>
> [2] Athalye, Anish, Nicholas Carlini, and David Wagner. "Obfuscated gradients give a false sense of security: Circumventing defenses to adversarial examples." arXiv preprint arXiv:1802.00420 (2018).
>
> [3] Tramer, Florian, et al. "On adaptive attacks to adversarial example defenses." arXiv preprint arXiv:2002.08347 (2020).
>
> [4] Carlini, Nicholas, et al. "On evaluating adversarial robustness." arXiv preprint arXiv:1902.06705 (2019).

---

> ### Author Response · Authors · 2020-11-14
> **Response to Reviewer3's questions (Part 2)**
>
> [**Overhead**] We also thank the reviewer for this valuable suggestion. To address the concerns, we profile 1) the number of parameters, 2) the inference time, and 3) the GPU memory usage (batch size = 8) on the PointNet backbone, as shown in the following table. Specifically, the inference time is the averaged time on the validation set (2468 objects) from ModelNet40, and all experiments are performed on a single 8GB RTX2080 GPU.
>
> |      | MAX | Median | Sum | PMA | ATT | ATT-Gate | FSPool | SoftPool | DeepSym |
> | : ------ | : ------ | : ----- | : ---- | : ---- | : ---- | : ---- | : ---- | : ---- | : ---- |
> | Inference time (ms) &nbsp; &nbsp; | 2.21 | 2.44 | 2.23 | 2.10 | 2.71 | 3.07 | 2.89 | 2.85 | 3.10 |
> | Trainable parameters &nbsp; &nbsp;| 815,336 &nbsp; | 815,336 &nbsp; | 815,336 &nbsp; &nbsp;| 652,136 &nbsp; &nbsp;| 1,340,649 &nbsp; &nbsp;| 1,865,962 &nbsp; &nbsp;| 1,863,912 &nbsp; &nbsp;| 355,328 &nbsp; &nbsp;| 1,411,563 &nbsp; &nbsp;|
> | GPU memory (MB) &nbsp; &nbsp; | 989 | 989 | 989 | 981 | 1980 | 2013 | 1005 | 725 | 2013 |
>
> As shown, DeepSym indeed introduces more computation by leveraging the shared MLP. However, we think the overhead is relatively small and acceptable, compared to the massive improvements on the adversarial robustness. To further have a lateral comparison, point cloud classification backbones are much more light-weight than image classification models. For example, the basic VGG-16 takes 50+ms to do the inference in our GPU. Therefore, we consider the overhead introduced by DeepSym acceptable.
> Moreover, DeepSym actually does not introduce more trainable parameters than FSPool, as mentioned in our first response. We can do a quick math here, FSPool will introduce $n \times d_m = 1024 \times 1024 = 1,048,576$ parameters. However, DeepSym adds $1024 \times 512 + 512 \times 128 + 128 \times 32 + 32 \times 8 + 8 \times 1 = 594,184$ additional parameters in the dense layers. The reason that SoftPool and PMA based PointNets have fewer trainable parameters than others is that both of them need $d_m$ to be relatively small. We are more than willing to add a section in the paper to discuss and clarify the overhead of DeepSym.
>
> We thank the reviewer again for the insightful comments, and we are happy to answer any future questions. We hope you might reconsider your rating based on our response.

---

> ### Author Response · Authors · 2020-11-22
> **Follow-up**
>
> Dear reviewer,
>
> We thank you again for the insightful feedback. We believe your comments raised have been addressed in our response and updated draft. We'd like to kindly and respectfully follow up and clarify any remaining issues. Your suggestions have been very important and valuable for us to improve the work!
>
> Thank you,
>
> Paper556 Authors

---

### Official Review · AnonReviewer1 · 2020-10-28
**Interesting direction, limited novelty and performance improvement**

**Rating:** 6
**Confidence:** 4

**Review:**

This paper studies adversarial robustness of point cloud classification models. In particular, this paper analyzes the effects of pooling layers and conducts extensive ablation studies. In addition, this paper proposes a DeepSym operation, which is built on top of both the sorting-based pooling and the parameterized pooling. The paper shows empirical improvements under different types of attacks. I summarize the pros and cons as follows.

Pros:
1.  Robustness of point cloud classification is an important problem to study. Compared to existing works on robustness, point cloud models have unique architectures, e.g, the pooling operation, which could be an important factor to the final performance. This paper studies an important problem.
2. This paper conducts multiple experiments of different pooling layers under different types of attacks.
3. Empirically, this paper shows improvements under different attacks.

Cons:
1. My biggest concern of this paper is its novelty. The sorting-based pooling and parameterized pooling layers are not new and have been studied extensively in existing papers.  FSPool proposed the sorting-based pooling while the parameterized pool layers have been proposed in multiple papers (e.g, PMA). Can the authors state the novelty/difference between this proposed method and existing ones?
2. The proposed pooling layers introduce additional parameters compared to sorting-based pooling layers, this makes the comparison less fair. Can the author talk about the number of parameters introduced by the proposed pooling layers. In particular, I'd like to see how much computational overhead introduced by the additional parameters? Is it possible to measure the inference time and compare to existing sorting-based methods (e.g, FSPool)?
3. The proposed pooling layers shows performance decrease when there is no adversarial attack. I'd like to see more discussion regarding this. Is this method specific to adversarial defense? Or this paper is proposing a general pooling layer?

---

> ### Author Response · Authors · 2020-11-14
> **Response to Reviewer1's questions (Part 1)**
>
> We thank the reviewer for the positive feedback and constructive comments. Below we respond to each question of the reviewer:
>
> [**Novelty**] First, we agree with the reviewer that parametric pooling operations have been studied in the literature. However, no existing works study their applications on the adversarial robustness of point cloud classification, which has important security implications in application domains such as autonomous driving. Previous work FSPool targets on point cloud autoencoders, and PMA was proposed for set structure learning. Therefore, one of our contributions is to unveil and benchmark the robustness of different pooling operations as the symmetric function in point cloud learning.  Specifically, one part of the novelty of our paper comes from the identification that the symmetric function significantly affects the robustness of point cloud classification. In summary, the contribution of this paper is three-fold. The adaptive attacks on state-of-the-art defenses first motivate us to study the adversarial training performance on point cloud classification. The success of DeepSym is based on the insights from the failure of fixed pooling operations and the comprehensive benchmarking of parametric pooling operations. Thus, DeepSym is actually built upon multiple valuable findings from the first part of our paper, and we believe the integrated contributions are strong enough to make our paper stand out.
>
> Second, we believe DeepSym itself has novelty. The key difference between DeepSym and others is the usage of a shared MLP. The original FSPool applies $n \times d_m$ learnable weights to the sorted feature map, which is a large number of parameters to train, and the linear transformation also lacks expressiveness. This design further limits FSPool to go deeper since the number of parameters will exponentially increase. The design of SoftPool is very complicated so that it requires the $d_m$ to be a small number to make the training tractable. Thus, SoftPool cannot achieve universality, which requires $d_m >= n$ \[1\]. In contrast, DeepSym has fairly good expressiveness since the deep networks are shown to have much more capacity, which also explains why DeepSym achieves good performance in both nominal and adversarial accuracy. Since the MLP is shared by different feature dimensions, the complexity increase of DeepSym is also well balanced. We will quantitatively show it in our second response to address the overhead concerns. Besides, DeepSym can also realize universality at the same time since it accepts $d_m >= n$. Moreover, DeepSym is by design flexible with the number of pooled features from each dimension. In the paper, we only allow DeepSym to output a single feature for a fair comparison with others. However, it is hard for other pooling operations to have this ability. For example, it requires a linear complexity increase for FSPool to enable this ability. However, DeepSym can achieve it with barely any complexity increases.
>
> Third, we also believe that the performance improvement should be a more important factor than the design complexity. The robustness enhancement by DeepSym is indeed significant. In the area of adversarial machine learning, various ideas that claimed to dramatically improve the adversarial robustness have been eventually demonstrated to be **gradient hiding** based methods\[2\], like DUP-Net and GvG-PointNet++. More papers with new ideas are broken by simple adaptive attacks very recently \[3\]. Such “novel” ideas actually cannot provide real adversarial robustness. As suggested in “On Evaluating Adversarial Robustness” by Carlini et al. \[4\], it is important for one paper to conduct correct evaluations rather than novel ideas without justification. Adversarial training-based methods are widely believed as the most effective way to mitigate the adversarial attacks. Thus, we believe that as the first study on the **real** robustness of point cloud classification, an absolute 28.5% ($2.6 \times$) and 6.5% improvement by DeepSym compared to Max and FSPool respectively should be considered significant.
>
> [1] Edward Wagstaff, Fabian B Fuchs, Martin Engelcke, Ingmar Posner, and Michael Osborne. On the limitations of representing functions on sets. arXiv preprint arXiv:1901.09006, 2019.
>
> [2] Athalye, Anish, Nicholas Carlini, and David Wagner. "Obfuscated gradients give a false sense of security: Circumventing defenses to adversarial examples." arXiv preprint arXiv:1802.00420 (2018).
>
> [3] Tramer, Florian, et al. "On adaptive attacks to adversarial example defenses." arXiv preprint arXiv:2002.08347 (2020).
>
> [4] Carlini, Nicholas, et al. "On evaluating adversarial robustness." arXiv preprint arXiv:1902.06705 (2019).

---

> ### Author Response · Authors · 2020-11-14
> **Response to Reviewer1's questions (Part 2)**
>
> [**Overhead**] We appreciate the reviewer for this valuable suggestion. To address the concerns, we profile 1) the number of parameters, 2) the inference time, and 3) the GPU memory usage (batch size = 8) on the PointNet backbone, as shown in the following table. Specifically, the inference time is the averaged time on the validation set (2468 objects) from ModelNet40, and all experiments are performed on a single 8GB RTX2080 GPU.
>
> |      | MAX | Median | Sum | PMA | ATT | ATT-Gate | FSPool | SoftPool | DeepSym |
> | : ------ | : ------ | : ----- | : ---- | : ---- | : ---- | : ---- | : ---- | : ---- | : ---- |
> | Inference time (ms) &nbsp; &nbsp; | 2.21 | 2.44 | 2.23 | 2.10 | 2.71 | 3.07 | 2.89 | 2.85 | 3.10 |
> | Trainable parameters &nbsp; &nbsp;| 815,336 &nbsp; | 815,336 &nbsp; | 815,336 &nbsp; &nbsp;| 652,136 &nbsp; &nbsp;| 1,340,649 &nbsp; &nbsp;| 1,865,962 &nbsp; &nbsp;| 1,863,912 &nbsp; &nbsp;| 355,328 &nbsp; &nbsp;| 1,411,563 &nbsp; &nbsp;|
> | GPU memory (MB) &nbsp; &nbsp; | 989 | 989 | 989 | 981 | 1980 | 2013 | 1005 | 725 | 2013 |
>
> As shown, DeepSym indeed introduces more computation by leveraging the shared MLP. However, we think the overhead is relatively small and acceptable, compared to the massive improvements on the adversarial robustness. To further have a lateral comparison, point cloud classification backbones are much more light-weight than image classification models. For example, the basic VGG-16 architecture containing 138 million parameters takes 50+ms to do the inference in our GPU. Therefore, we consider the overhead introduced by DeepSym acceptable.
> Moreover, DeepSym actually does not introduce more trainable parameters than FSPool, as mentioned in our first response. We can do a quick math here, FSPool will introduce $n \times d_m = 1024 \times 1024 = 1,048,576$ parameters. However, DeepSym adds $1024 \times 512 + 512 \times 128 + 128 \times 32 + 32 \times 8 + 8 \times 1 = 594,184$ additional parameters in the dense layers. The reason that SoftPool and PMA based PointNets have fewer trainable parameters than others is that both of them need $d_m$ to be relatively small. We are more than willing to add a section in the paper to discuss and clarify the overhead of DeepSym.
>
> [**Performance on the clean data**] We believe DeepSym can serve as a general pooling operation in DNN targeting point cloud processing and set structure learning. We agree with the reviewer’s assessment that the nominal accuracy (accuracy of clean data) will drop after adversarial training. However, this is a normal phenomenon which has been acknowledged by many top-tier papers [1,2,3] in the adversarial machine learning area. The reason behind this is that the training is based on adversarial data which has a distribution shift from the clean point cloud, and [4] shows that there is an intrinsic trade-off between accuracy and robustness. Moreover, DeepSym actually achieves the second-highest nominal accuracy, as shown in Figure 5 (Median performs the best in nominal accuracy but the worst in adversarial robustness). Therefore, DeepSym in fact realizes the best balance between nominal accuracy and adversarial robustness.
>
> We thank the reviewer again for the insightful comments, and we are happy to answer any future questions. We hope you might reconsider your rating based on our response.
>
> [1] Madry, Aleksander, et al. "Towards deep learning models resistant to adversarial attacks." arXiv preprint arXiv:1706.06083 (2017).
>
> [2] Xie, Cihang, and Alan Yuille. "Intriguing properties of adversarial training at scale." arXiv preprint arXiv:1906.03787 (2019).
>
> [3] Kurakin, A., Goodfellow, I., & Bengio, S. (2016). Adversarial machine learning at scale. arXiv preprint arXiv:1611.01236.
>
> [4] Tsipras, D., Santurkar, S., Engstrom, L., Turner, A., & Madry, A. (2018). Robustness may be at odds with accuracy. arXiv preprint arXiv:1805.12152.

---

> ### Author Response · Authors · 2020-11-22
> **Follow-up**
>
> Dear reviewer,
>
> We thank you again for the insightful feedback. We believe your comments raised have been addressed in our response and updated draft. We'd like to kindly and respectfully follow up and clarify any remaining issues. Your suggestions have been very important and valuable for us to improve the work!
>
> Thank you,
>
> Paper556 Authors

---

### Official Review · AnonReviewer4 · 2020-10-29
**Interesting work with good experiments**

**Rating:** 7
**Confidence:** 3

**Review:**

The paper addresses the problem of adversarial robustness in 3D point cloud representations. It claims that two of the previous defense designs do not prevent adaptive attacks. The authors then propose to use adversarial training (AT) to improve the robustness. It claims that the standard MAX pooling operation within PointNet-derivates contribute to the weaknesses. It then proposes a new pooling operation that improves the robustness under AT.

The paper claims contributions in:

-- Demonstration that current defenses do not provide real robustness.

-- MAX and SUM pooling strategies weaken models trained under AT.

-- A new sorting-based pooling operation, DeepSym, that works the best with AT against adaptive attacks.

I agree with the authors' assessment of the paper.

Strengths:

-- Interesting insights: As promised, the authors applied adaptive attacks against recent works that aimed to improve robustness. The experiments are clear and discussions make sense. The discovery that sorting-based pooling layers are more robust under AT is also insightful.

-- Strong experimental results: On the one task of ModelNet40, the authors showed significant improvement using three methods when using AT, and further improvements when using the proposed DeepSym layer.

Neutral:

-- Novelty: The majority of claimed novelties are straightforward applications: 1) application of AT on points, 2) usage of attention or sorting-based layer to improve robustness. The proposal of DeepSym is somewhat novel, but not sure if sufficient for a publication in this venue.

Weaknesses:

-- There is only one task: All experiments were done on ModelNet40. Having a second dataset would be nice. Or even some toy examples.

Clarification:

-- I imagine that the adaptive attacks are white box attacks? I couldn't find in the paper if white or black attack....

Conclusion:
The paper provides interesting insights and strong experimental results. Although the novelty is slightly lackluster, I would be happy to see this paper at the conference.

---

> ### Author Response · Authors · 2020-11-14
> **Response to Reviewer4's questions**
>
> We thank the reviewer for the positive feedback and constructive comments. Below we respond to each question of the reviewer:
>
> [**Dataset issue**] We appreciate the reviewer for this valuable suggestion. ModelNet40 is the most popular dataset for benchmarking the point cloud classification task. In Section 5.1, we utilize ModelNet10, which is a smaller dataset to assess the adversarial robustness. We find one ICCV’19 paper releasing a point cloud dataset called ScanObjectNN \[1\], and we are trying our best to integrate this dataset into our evaluation. We will update the draft once the results are ready.
>
> [**Adaptive attack**] Yes, the designed adaptive attacks are both white-box attacks. We will explicitly clarify this in the updated version of our paper.
>
> Thanks again for your positive review, and we are happy to answer any future questions.
>
> [1] Uy, Mikaela Angelina, et al. "Revisiting point cloud classification: A new benchmark dataset and classification model on real-world data." Proceedings of the IEEE International Conference on Computer Vision. 2019.

---

### Official Review · AnonReviewer2 · 2020-11-03
**This paper analyzes the robustness of 3D point cloud classification, while there are some concerns need to be figured out.**

**Rating:** 5
**Confidence:** 4

**Review:**

This paper first points out the drawback of state-of-the-art defenses for 3D point cloud classification, via designing adaptive attack. And then the author analyzes the effect of adversarial training on this task. Based on such analysis, the authors propose a deep symmetric pooling operation which can enhance the adversarial robustness for adversarial training.

The presentation of the paper is clear and easy to follow. And there are extensive experiments and sufficient experimental analysis to support the claim of this paper. However, there are still some concerns that need to be solved.

1. The authors utilize the white-box attack to analyze the robustness of different pooling operations, while they should also compare the defense effect of various pooling operations under black-box attack.
2. In table 2, although the adaptive PGD attack has high success rates, the attack with L_{gather} has higher performance in some cases. What is the intrinsic reason for its superiority?
3. In the experiments of adversarial training, the authors propose to select PGD for the training. However, the authors evaluate the performance of trained models with PGD only, e.g. the results in Tables 3 and 4.  It is necessary for the authors to evaluate the results under various attack types.

---

> ### Author Response · Authors · 2020-11-15
> **Response to Reviewer2's questions**
>
> We thank the reviewer for the positive feedback and constructive comments. Below we respond to each question of the reviewer:
>
> [**Black-box attacks**] We appreciate the reviewer for this valuable suggestion. The original reason that we leverage white-box attacks to assess the adversarial robustness of different backbones is that white-box attacks are stronger than black-box attacks. We are willing to add black-box attacks on PointNet to make our evaluation more comprehensive. We are currently working hard on it. We will update this comment and the paper once the experiment results come out.
>
> [**Results in Table 2**] We thank the reviewer for carefully checking the detailed evaluation of our adaptive attack. The intuition behind the attacks on $L_{gather}$ is that such an attack will fool the gather vector predictions so that most features will be masked out. As shown in Table 2 and indicated by the reviewer, we also observe that $L$-inf norm-based PGD attack is more effective on $L_{gather}$ since $L$-inf norm perturbations assign the same adversarial budget to each point, which can easily impact a large number of gather vector predictions. However, it is hard for the $L$-2 norm-based PGD attack to influence all the gather vector predictions because it prefers to perturb the key points rather than the whole point set. In an extreme case, the $L$-2 norm-based PGD attack might make a single gather prediction very far away from the ground-truth, but the rest gather vectors are still sufficient to make the classification correct. However, since there is an absolute boundary for each point, the adversary instead prefers to perturb most gather vector predictions under $L$-inf norm distance.
> In the original GvG-PointNet++ paper, the authors did not evaluate $L$-inf norm-based attacks, and we believe it is an inappropriate evaluation, as suggested in “On Evaluating Adversarial Robustness” by Carlini et al. [1].
>
> [**Additional white-box attacks**] We leverage the PGD attack since it tends to be a universal first-order attack method [2]. Although we select PGD attacks in both training and inference phases, the inference-time PGD attack is much stronger with 200 steps which could find harsher adversarial examples. We definitely agree with the reviewer to add more white-box attacks on PointNet in our evaluation. Specifically, we select the reputational white-box attack methods: FGSM [3], BIM [4], and MIM [5]. FGSM is a single-step method, and BIM and MIM are iterative attack methods. We also choose 200 steps for the iterative methods in these experiments, and all the experiments are performed on the PGD-7 trained PointNet under $\epsilon = 0.05$. We will elaborate on the detailed setups in our revised paper. The following table shows the results of adversarial accuracy:
>
> |      | MAX | Median | Sum | PMA | ATT | ATT-Gate | FSPool | SoftPool | DeepSym |
> | : ------ | : ------ | : ----- | : ---- | : ---- | : ---- | : ---- | : ---- | : ---- | : ---- |
> | FGSM (%) &nbsp; &nbsp; | 72.8 | **77.6** | 44.4 | 47.2 | 43.1 | 43.9 | 61.3 | 62.1 | 61.4 |
> | BIM (%) &nbsp; &nbsp; | 24.3 | 23.3 | 33.5 | 31.9 | 33.1 | 34.2 | 45.4 | 47.6 | **52.5** |
> | MIM (%) &nbsp; &nbsp; | 23.5 | 14.5 | 37.4 | 30.1 | 35.0 | 33.9 | 48.0 | 45.1 | **55.4** |
>
> As shown, PointNet with DeepSym still performs the best in adversarial accuracy under BIM and MIM. DeepSym also expands the performance lead compared to both FSPool and SoftPool under these two iterative attacks. The reason that Max and Median pooling perform better under FGSM is that FGSM is much weaker than the others to find adversarial examples. Therefore the number of FGSM is not as representative as others.
>
> We believe that it is imperative to correctly study the adversarial robustness of point cloud classification, which has important security implications in application domains such as autonomous driving. A timely and comprehensive study on the robustness of the classification task could also benefit future research.
>
> We thank you again for the insightful review, and we are happy to answer any future questions. We hope you might reconsider your rating based on our response.
>
> [1] Carlini, Nicholas, et al. "On evaluating adversarial robustness." arXiv preprint arXiv:1902.06705 (2019).
>
> [2] Madry, Aleksander, et al. "Towards deep learning models resistant to adversarial attacks." arXiv preprint arXiv:1706.06083 (2017).
>
> [3] Szegedy, C., Zaremba, W., Sutskever, I., Bruna, J., Erhan, D., Goodfellow, I., & Fergus, R. (2013). Intriguing properties of neural networks. arXiv preprint arXiv:1312.6199.
>
> [4] Kurakin, A., Goodfellow, I., & Bengio, S. (2016). Adversarial examples in the physical world. arXiv preprint arXiv:1607.02533.
>
> [5] Dong, Y., Liao, F., Pang, T., Su, H., Zhu, J., Hu, X., & Li, J. (2018). Boosting adversarial attacks with momentum. In Proceedings of the IEEE conference on computer vision and pattern recognition (pp. 9185-9193).

---

> > ### Author Response · Authors · 2020-11-19
> > **Black-box attack evaluation results**
> >
> > We have finished the implementations and evaluations of black-box adversarial attacks on PGD-7 trained PointNet.  We choose two score-based methods: SPSA [1] and NES [2], and a decision-based evolution attack [3]. We still select $\epsilon = 0.05$ and allow 2000 queries (due to time and computational resource constraints) to find each adversarial example. The detailed setups will be elaborated in our revised paper. We showcase the results in the following table:
> >
> > |      | MAX | Median | Sum | ATT | ATT-Gate | PMA | FSPool | SoftPool | DeepSym |
> > | : ------ | : ------ | : ----- | : ---- | : ---- | : ---- | : ---- | : ---- | : ---- | : ---- |
> > | SPSA (%) &nbsp; &nbsp; | 69.2 | 71.1 | 65.3 | 68.1 | 70.2 | 67.2 | **72.8** | 69.2 | 72.4 |
> > | NES (%) &nbsp; &nbsp; | 67.1 | 65.2 | 62.3 | 64.8 | 65.9 | 64.1 | 71.9 | 68.5 | **72.1** |
> > | Evolution (%) &nbsp; &nbsp; | 53.4 | 57.8 | 52.7 | 55.9 | 55.8 | 53.4 | 69.9 | 70.0 | **73.1** |
> >
> > As shown and expected, the black-box attacks are not as effective as white-box attacks to break the robustness of PGD-7 trained models. DeepSym also performs well under those black-box adversarial attacks. This also validates that adversarially trained models are able to offer real robustness rather than gradient obfuscation.
> >
> > We believe that we have addressed all the raised concerns, and we are happy to answer any follow-up questions. We hope the reviewer might reconsider the rating based on our response.
> >
> > [1] Jonathan Uesato, Brendan O’Donoghue, Aaron van den Oord, and Pushmeet Kohli. Adversarial riskand the dangers of evaluating against weak attacks.arXiv preprint arXiv:1802.05666, 2018.
> >
> > [2] Andrew Ilyas, Logan Engstrom, Anish Athalye, and Jessy Lin.  Black-box adversarial attacks withlimited queries and information.arXiv preprint arXiv:1804.08598, 2018.
> >
> > [3] Yinpeng Dong, Qi-An Fu, Xiao Yang, Tianyu Pang, Hang Su, Zihao Xiao, and Jun Zhu. Benchmark-ing adversarial robustness on image classification.  InProceedings of the IEEE/CVF Conferenceon Computer Vision and Pattern Recognition, pp. 321–331, 2020b.

---

> ### Author Response · Authors · 2020-11-22
> **Follow-up**
>
> Dear reviewer,
>
> We thank you again for the insightful feedback. We believe your comments raised have been addressed in our response and updated draft. We'd like to kindly and respectfully follow up and clarify any remaining issues. Your suggestions have been very important and valuable for us to improve the work!
>
> Thank you,
>
> Paper556 Authors

---

### Author Response · Authors · 2020-11-21
**A general response to all reviewers**

Dear Reviewers,

We would like to thank all reviewers for the time and effort spent reviewing our paper. Your constructive and insightful comments are valuable to help us refine and improve our paper. We have responded to each reviewer in detail so far, and we believe that we have addressed all the raised questions.

We have uploaded a revised version of our paper, and we summarize the revision here for the convenience of further reviewing our paper.

1. \[**Novelty**\] We have updated Section 5 to clarify the novelty of DeepSym. We also responded in detail to Reviewer 1 & 3 about the key differences between DeepSym and other pooling operations.

2. \[**Evaluation on other attacks**\] We have updated Section 5.1 to further evaluate 6 kinds of attacks on adversarially trained PointNet with different pooling operations (3 white-box attacks and 3 black-box attacks). The results show that PointNet with DeepSym still achieves the best adversarial accuracy under almost all the attacks, which also strengthens our paper.

3. \[**Dataset**\] We have updated Appendix C.2 to show the evaluation of different pooling operations on ScanObjectNN, a new real-world point cloud dataset. We demonstrate that models with DeepSym still achieve the best results on this dataset.

4. \[**Overhead**\] We have updated Section 5.1 to discuss the overhead of DeepSym. As we responded to Reviewer 1 & 3, we agree that DeepSym will bring extra computational overhead. However, we believe that the overhead is relatively small and acceptable, compared to the large improvements on the adversarial robustness.

5. \[**Clarification**\] We clarify in Section 1 that our designed adaptive attacks belong to the white-box attack. We also add more explanations on the results in Table 2 (i.e., attacks on $L_{gather}$ are more effective than attacks on $L_{adv}$) in the last paragraph of Section 3.2.

We believe that it is imperative to correctly study the adversarial robustness of point cloud classification, which has important security implications in application domains such as autonomous driving. Therefore, we hope the reviewers might reconsider their ratings. We are happy to answer any future questions.

Thank you!

Paper556 Authors

---

### Decision · Program_Chairs · 2021-01-07
**Final Decision**

**Decision:**

Reject

**Comment:**

The authors develop novel adaptive adversarial attacks for 3D Point Cloud Classification tasks. They show that many existing defenses are broken by develop a novel pooling operation, DeepSym, and demonstrate that using this they can achieve significant improvements in adversarial robustness of 3D Point Cloud Classification.

All reviewers agreed that the paper makes interesting contributions and that the setting of 3D point cloud classification is interesting from a security perspective. The shared concern of the reviewers was around novelty. This was addressed in the rebuttal to some extent, but there remained some lingering questions that made this paper borderline and unfortunately, the program committee had to decide to reject it.

I would urge the authors to revise their manuscript to clarify the novelty relative to prior work (particular those that use similar pooling operations) and resubmit to a future venue.